# Switch-like Arp2/3 activation upon WASP and WIP recruitment to an apparent threshold level by multivalent linker proteins in vivo

Yidi Sun, Nicole T Leong, Tommy Jiang, Astou Tangara, Xavier Darzacq, David G Drubin*

Department of Molecular and Cell Biology, University of California, Berkeley, Berkeley, United States

**Abstract** Actin-related protein 2/3 (Arp2/3) complex activation by nucleation promoting factors (NPFs) such as WASP, plays an important role in many actin-mediated cellular processes. In yeast, Arp2/3-mediated actin filament assembly drives endocytic membrane invagination and vesicle scission. Here we used genetics and quantitative live-cell imaging to probe the mechanisms that concentrate NPFs at endocytic sites, and to investigate how NPFs regulate actin assembly onset. Our results demonstrate that SH3 (Src homology 3) domain-PRM (proline-rich motif) interactions involving multivalent linker proteins play central roles in concentrating NPFs at endocytic sites. Quantitative imaging suggested that productive actin assembly initiation is tightly coupled to accumulation of threshold levels of WASP and WIP, but not to recruitment kinetics or release of autoinhibition. These studies provide evidence that WASP and WIP play central roles in establishment of a robust multivalent SH3 domain-PRM network in vivo, giving actin assembly onset at endocytic sites a switch-like behavior.

*For correspondence: drubin@berkeley.edu

**Competing interests:** The authors declare that no competing interests exist.

## Introduction

Nucleation-promoting factors (NPFs) activate the actin-related protein 2/3 (Arp2/3) complex to assemble actin filaments, which are important in many cellular processes, such as morphogenesis, cell motility and endocytosis (*Badour et al., 2003*; *Campellone and Welch, 2010*; *Rottner et al., 2010*). Moreover, NPF-mediated actin nucleation can be co-opted by pathogens to mediate infection and spread (*Welch and Way, 2013*). How NPF activity is regulated in a spatiotemporal manner in cells is an important open question. In addition to VCA motifs (composed of a Verprolin homology domain, Central hydrophobic region, and Acidic region) that directly stimulate Arp2/3 complex-mediated actin nucleation, NPFs often contain additional functional elements that mediate interactions with other proteins (*Padrick and Rosen, 2010*; *Takenawa and Suetsugu, 2007*). For example, the EVH1 domain (Ena/VASP homology) of N-WASP (Neural Wiskott-Aldrich Syndrome protein) interacts with WIP (WASP-interacting protein) (*Donnelly et al., 2013*; *Rivera et al., 2004*; *Tehrani et al., 2007*). In addition, most WASP family proteins, which are the best-studied NPFs, also contain a large proline-rich domain (PRD) that contains multiple Src homology 3 (SH3)-binding proline-rich motifs (PRMs). Previous studies demonstrated that dozens of SH3-domain containing ligands can bind to WASP PRMs, and that some enhance WASP NPF activity (*Donnelly et al., 2013*; *Padrick and Rosen, 2010*; *Rivera et al., 2004*; *Tehrani et al., 2007*). Multivalent PRM-SH3 domain interactions between N-WASP and its ligands enable proteins to form higher order complexes in vitro on artificial membranes and undergo a phase separation, producing micrometer-size clusters

**eLife digest** Actin is one of the most abundant proteins in yeast, mammalian and other eukaryotic cells. It assembles into long chains known as filaments that the cell uses to generate forces for various purposes. For example, actin filaments are needed to pull part of the membrane surrounding the cell inwards to bring molecules from the external environment into the cell by a process called endocytosis.

In yeast, a member of the WASP family of proteins promotes the assembly of actin filaments around the site where endocytosis will occur. To achieve this, WASP interacts with several other proteins including WIP and myosin, a motor protein that moves along actin filaments to generate mechanical forces. However, it was not clear how these proteins work together to trigger actin filaments to assemble at the right place and time.

Sun et al. addressed this question by studying yeast cells with genetic mutations affecting one or more of these proteins. The experiments show that WASP, myosin and WIP are recruited to sites where endocytosis is about to occur through specific interactions with other proteins. For example, a region of WASP known as the proline-rich domain can bind to proteins that contain an "SH3" domain. WASP and WIP arrive first, stimulating actin to assemble in an "all and nothing" manner and attracting myosin to the actin. Further experiments indicate that WASP and WIP need to reach a threshold level before actin starts to assemble.

The findings of Sun et al. suggest that WASP and WIP play key roles in establishing the network of proteins needed for actin filaments to assemble during endocytosis. These proteins are needed for many other processes in yeast and other cells, including mammalian cells. Therefore, the next steps will be to investigate whether WASP and WIP use the same mechanism to operate in other situations.

(*Banjade and Rosen, 2014*; *Li et al., 2012*). Interestingly, such clusters robustly trigger Arp2/3-mediated actin assembly (*Banjade and Rosen, 2014*). However, how N-WASP and WIP become highly concentrated locally to trigger actin assembly in cells is not well understood.

In budding yeast, all known NPFs and over 50 other proteins are organized into plasma membrane-associated patches, which are clathrin-mediated endocytosis (CME) sites (*Kaksonen et al., 2005*). Extensive efforts have been made to determine the functions of these NPFs and to identify their interacting partners both in vitro and in vivo (*Boettner et al., 2011*; *Goode et al., 2015*). In addition, a very detailed pathway has been elucidated in which each endocytic protein is recruited to endocytic sites in a predictable order and with predictable timing (*Figure 1A*)(*Lu et al., 2016*). The two most important NPFs, yeast WASP (Las17) and type I myosin (Myo3 and Myo5), which form a complex with yeast WIP (Vrp1) and additional binding partners, can be grouped into a WASP-Myosin module (*Kaksonen et al., 2005*). The WASP-Myosin module stimulates actin filament assembly and provides motor activity, facilitating membrane invagination and vesicle scission (*Galletta et al., 2008*; *Lewellyn et al., 2015*; *Sirotkin et al., 2005*; *Soulard et al., 2002*; *Sun et al., 2006*). Individual components of the WASP-Myosin module are recruited to endocytic sites in a regular, timely order, in which Las17 appears first, followed by Vrp1, and finally Myo3/5, coinciding with the onset of robust actin filament assembly (*Sun et al., 2006*).

Recent studies revealed a condition in which WASP-Myosin module driven-actin assembly is uncoupled from cortical endocytic sites in live cells (*Bradford et al., 2015*; *Sun et al., 2015*). Multivalent linker proteins Pan1 and End3 appear to associate with each other constitutively (*Boeke et al., 2014*; *Sun et al., 2015*), and together with Sla1 (another multivalent linker protein that is recruited to endocytic sites by Pan1 and End3), likely provide equivalent functions to mammalian intersectins (ITSNs) (*Goode et al., 2015*). When Pan1 and End3 are eliminated from cells by an auxin-based degron method, endocytic sites still assemble at the cell cortex, but WASP-Myosin proteins associate with actin comet tails in the cytoplasm instead of with the endocytic sites (*Sun et al., 2015*). These findings indicate that the Pan1-End3-Sla1 complex plays essential roles in coupling actin assembly to endocytic sites. However, the Pan1-End3-Sla1 complex interacts with numerous

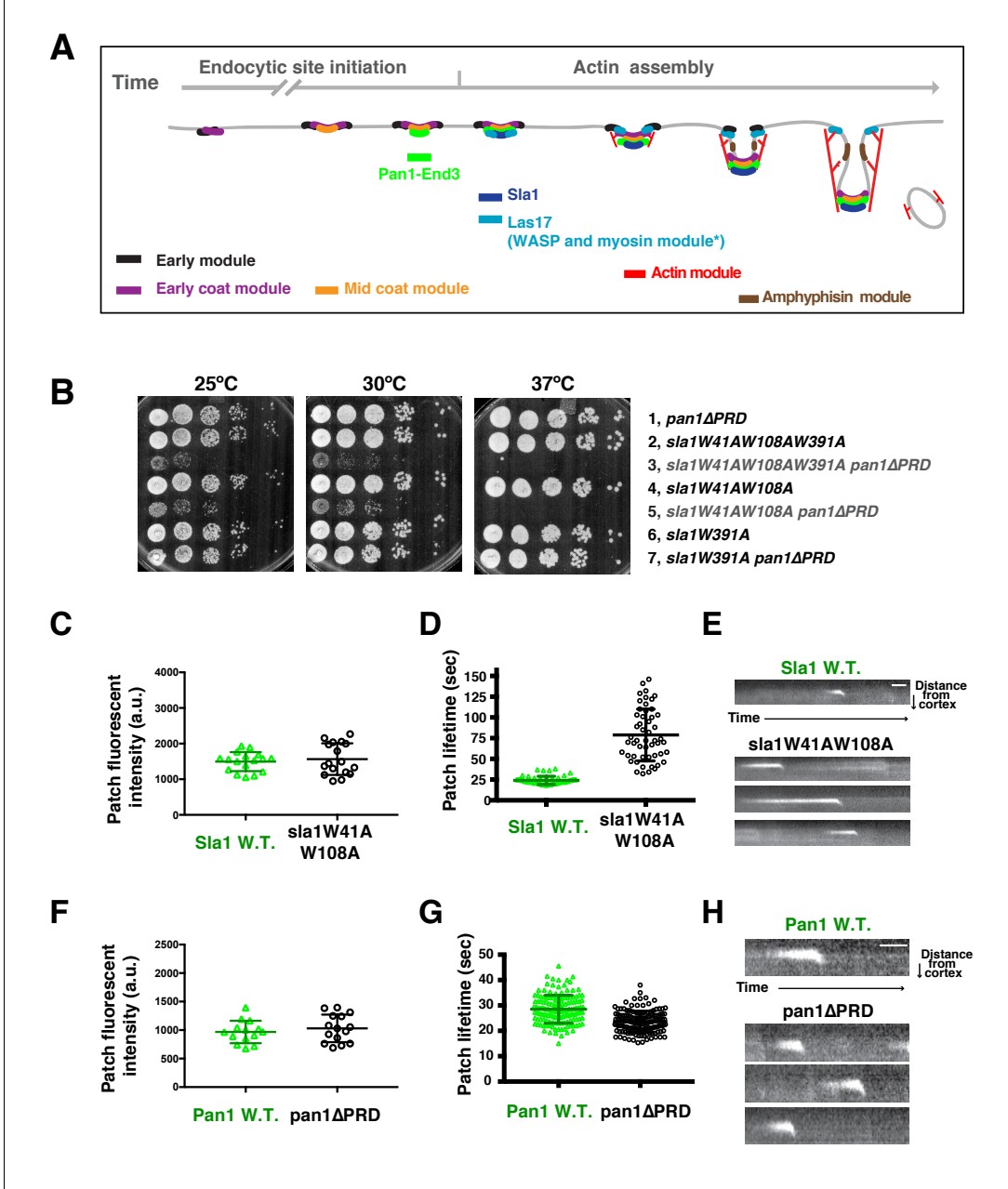

**Figure 1.** Two Sla1 SH3 domains and a Pan1 PRD domain share a crucial role for cell growth. (A) Spatial-temporal recruitment of endocytic proteins. Endocytic proteins are grouped into several modules (*Lu et al., 2016*) as indicated. Pan1 and End3 appear after the mid coat module proteins but slightly before Sla1 and Las17 appear (*Sun et al., 2015*). * Note that proteins of the WASP-Myosin module arrive at endocytic sites with different timing. Las17 arrives with a similar timing to Sla1, while the remaining components of the WASP-Myosin module arrive later (*Sun et al., 2006*). (B) Synthetic genetic interaction between *sla1W41AW108A* and *pan1ΔPRD*. Cell growth of indicated yeast strains was compared by spotting serial dilutions of liquid cultures on plates at 25°C, or 30°C or 37°C. (C-E) Analysis of sla1W41A-W108A-GFP dynamics. (F-H) Analysis of pan1ΔPRD-GFP dynamics. C and F, Maximum fluorescence intensity of GFP-tagged patch proteins at endocytic sites (also see *Figure 1—figure supplement 1D and E*). (D and G) Lifetime (mean ± SD) of GFP-tagged proteins. E and H, Radial kymograph representations (for explanation, please see *Figure 1—figure supplement 2*) of GFP-tagged proteins. The scale bars are 20 s.

The following figure supplements are available for figure 1:

**Figure supplement 1.** Characterization of *sla1W41AW108A* and *pan1ΔPRD* mutant cells.

**Figure supplement 2.** Flowchart for scheme used to generate radial kymograph of fluorescently labeled-proteins in a movie.

additional proteins throughout endocytosis. An important but challenging task is to identify the specific domain(s) and interactions required for linking actin assembly to endocytic sites.

In this study, to mechanistically understand how actin polymerization is coupled to endocytic sites, we identified the key interaction(s) required for concentrating the actin assembly machinery at endocytic sites and we further addressed how these interactions trigger a sudden burst of actin assembly. These results not only deepen our understanding of clathrin-mediated endocytosis, but also provide general mechanistic insights into spatiotemporal regulation of actin assembly in vivo.

## Results

### SH3 and proline-rich domains of two multivalent endocytic linker proteins provide crucial overlapping roles for cell growth

To explore the mechanism by which actin assembly is coupled to endocytic sites at a molecular level, we sought to identify the functional domains within the endocytic linker proteins that are responsible for the process. Previous studies revealed a synthetic lethal interaction between *sla1Δ* and *pan1ΔPRD*, in which the Pan1 C-terminal PRD (proline-rich domain) is truncated (*Barker et al., 2007*) (*Figure 1—figure supplement 1A*). However, the mechanistic basis for this lethality had not been explored. The Pan1 PRD has been shown to interact with yeast type 1 myosin (Myo3 and Myo5) SH3 domain and enhance Myo3/5-Vrp1 NPF activity in vitro (*Barker et al., 2007*). However, *sla1Δ* does not display a synthetic lethal interaction with *myo5CAΔmyo3Δ* (*Figure 1—figure supplement 1B*), in which the type 1 myosin NPF activity is abolished (*Sun et al., 2006*). Thus, *pan1ΔPRD* does not appear to cause synthetic lethality with *sla1Δ* by affecting Myo3/5-Vrp1 NPF activity. To gain insights into how Sla1 and Pan1 are related functionally, it was important to identify the Sla1 functions whose loss results in the synthetic lethal interaction with *pan1ΔPRD*. Therefore, we mutated various domains of Sla1 and crossed the mutants to *pan1ΔPRD*.

Strikingly, we found that two amino acid substitutions in Sla1 are sufficient to cause a synthetic genetic interaction with *pan1ΔPRD*. *sla1W41AW108A pan1ΔPRD* cells display severe growth defects at 25°C and 30°C, and are inviable at 37°C (*Figure 1B*). W41 and W108 are the conserved tryptophan residues in two SH3 (SRC homology 3) domains of Sla1 (*Figure 1—figure supplement 1A*) (*Rodal et al., 2003*). Point mutation of these sites abolishes the SH3 domain interactions with PRMs (*Rodal et al., 2003*). However, a point mutation (W391A) on the third SH3 domain of Sla1 did not show a synthetic interaction with *pan1ΔPRD* (*Figure 1B*). Thus, our results indicate that the first two SH3 domains of Sla1 and the PRD of Pan1 function in parallel to provide a crucial role for cell growth.

We next analyzed the *sla1W41AW108A* and *pan1ΔPRD* mutants separately. Immunoblotting of whole cell extracts showed that sla1W41AW108A-GFP is well expressed (*Figure 1—figure supplement 1C*). More importantly, sla1W41AW108A-GFP appeared in cortical patches that reached fluorescence intensity levels similar to wild-type Sla1-GFP patches (*Figure 1C* and *Figure 1—figure supplement 1D*). However, the sla1W41AW108A-GFP patch lifetimes were substantially longer and more variable compared to Sla1-GFP patches (78.8 ± 31.2 vs 24.1 ± 4.9, p<0.0001) (*Figure 1D*). Nevertheless, similar to the wild-type cells, sla1W41AW108A-GFP patches moved inward, off the cell cortex, at the end of their lifetime (*Figure 1E*), indicating that endocytic internalization still takes place in this mutant.

Immunoblotting showed that pan1ΔPRD-GFP is also expressed at levels similar to the wild-type protein (*Figure 1—figure supplement 1C*). In addition, cortical pan1ΔPRD-GFP patches reached fluorescence intensity levels similar to wild-type Pan1-GFP patches (*Figure 1F* and *Figure 1—figure supplement 1E*). Finally, pan1ΔPRD-GFP patch lifetimes were slightly shorter than Pan1-GFP lifetimes (23.6 ± 4.2 vs 28.5 ± 5.5, p<0.0001) (*Figure 1G*), and they were internalized at the end of their lifetime (*Figure 1H*).

The results described above establish that neither a point mutant of two Sla1 SH3 domains nor a PRD truncation mutant of Pan1 affects either protein's expression or cortical recruitment. Thus, the severe synthetic growth defect of an *sla1 W41AW108A pan1ΔPRD* double mutant is caused by the loss of specific functions rather than by the absence of the mutant proteins at endocytic sites. We next analyzed the *sla1 W41AW108A pan1ΔPRD* double mutant to determine how the Sla1 SH3 domains and the Pan1 PRD function in endocytosis.

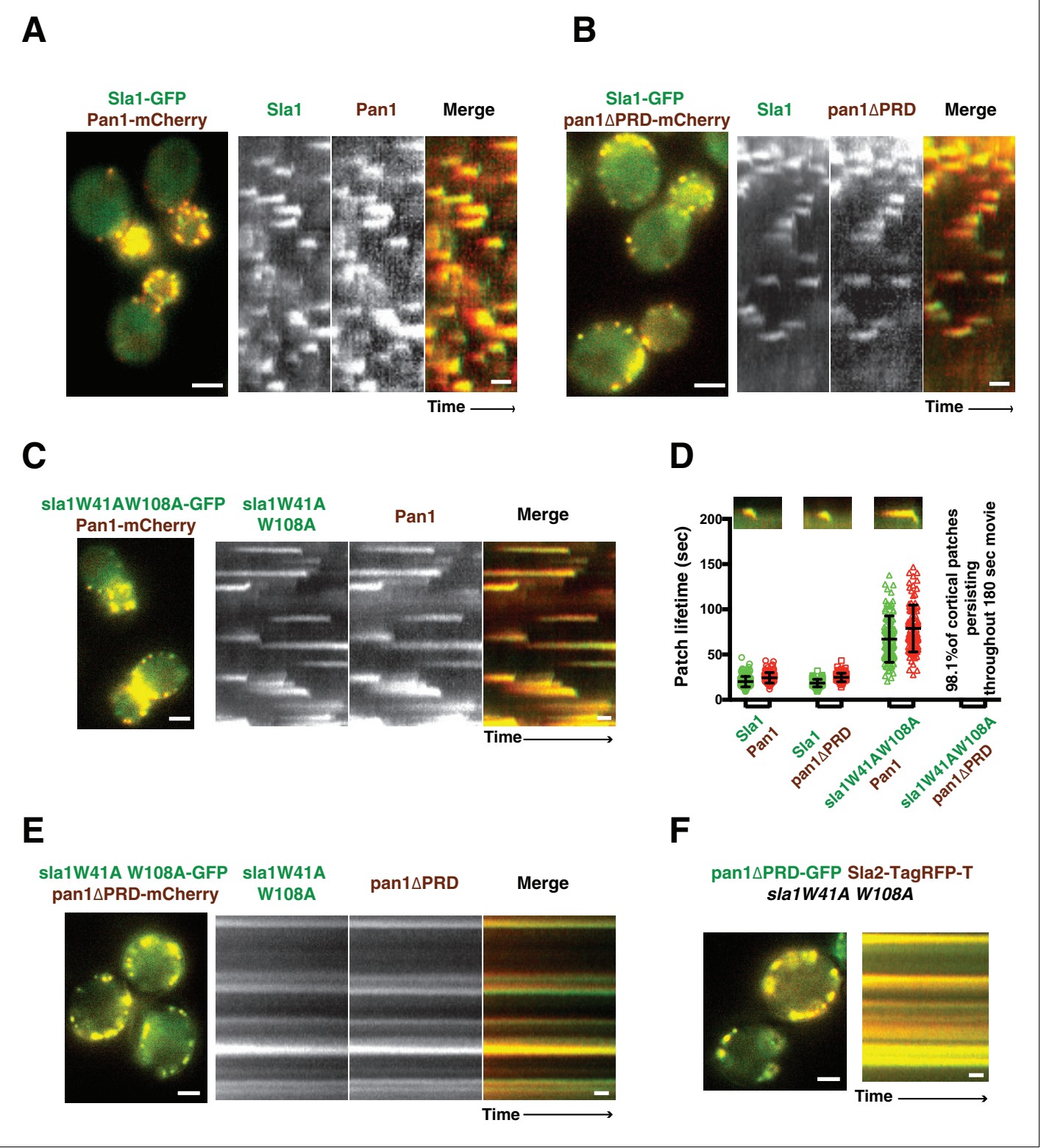

**Figure 2.** Endocytic internalization is defective in *sla1 W41AW108A pan1ΔPRD* mutant cells. (**A-C**) Single frames (left) from movies and circumferential kymograph representations (for the explanation, please see *Figure 2—figure supplement 1*) for GFP- and mCherry-tagged proteins. (**A**) *SLA1-GFP PAN1-mCherry* cells. (**B**) *SLA1-GFP pan1ΔPRD-mCherry* cells. (**C**) *sla1W41AW108A-GFP PAN1-mCherry* cells. (**D**) Lifetime (mean ±SD) and radial kymograph representations of GFP- and mCherry-tagged proteins for indicated strains. (**E**) Single frame (left) from a 3 min movie and circumferential kymograph representations of *sla1W41AW108A-GFP pan1ΔPRD-mCherry* cells. (**F**) Single frame (left) from 3 min movie and circumferential kymograph

*Figure 2 continued on next page*

*Figure 2 continued*

representation of pan1ΔPRD-GFP and Sla2-TagRFP-T in *sla1W41AW108A* cells. The scale bars on kymographs are 20 s. The scale bars on cell pictures are 2 µm.

The following figure supplements are available for figure 2:

**Figure supplement 1.** Flowchart for scheme used to generate circumferential kymograph of fluorescently labeled patch proteins on the cell cortex in a movie.

**Figure supplement 2.** Lucifer yellow (LY) uptake is defective in an *sla1W41AW108A pan1ΔPRD* strain.

## Endocytic internalization requires SH3 or proline-rich domains of multivalent endocytic linker proteins

We examined cortical patch behavior of GFP-tagged Sla1 (or sla1 mutant) and mCherry-tagged Pan1 (or pan1 mutant) in *sla1 W41AW108A* and/or *pan1ΔPRD* mutants.

Previous studies suggested that Sla1 is recruited to endocytic sites by Pan1 and End3 (*Sun et al., 2015*; *Tang et al., 2000*). Sla1 (or the sla1 mutant) appears slightly after Pan1 (or the pan1 mutant), and then internalizes in wild-type cells, *pan1ΔPRD* cells, and *sla1W41AW108A* cells (*Figure 2A–D*). Consistent with the results in *Figure 1D*, patch lifetimes in *sla1W41AW108A* cells are irregular and longer than in wild-type and *pan1ΔPRD* cells (*Figure 2D*). However, in the double mutants, sla1-W41AW108A and pan1ΔPRD colocalize as stable patches at the cell cortex (*Figure 2E*). Most cortical patches (98.1%, 216 cortical patches from 20 cells were examined) stay nonmotile during the entire 3 min movie, suggesting that endocytic internalization does not occur (*Figure 2D and E*). A Lucifer yellow uptake assay confirmed a severe endocytic defect (*Figure 2—figure supplement 2*). Moreover, in the double mutant cells, pan1ΔPRD colocalizes with Sla2 (*Figure 2F*), a signature protein for endocytic sites, and the yeast homologue of vertebrate HIP1R (*Engqvist-Goldstein et al., 2001*). These results indicate that the nonmotile cortical sla1W41AW108A/pan1ΔPRD patches are nonproductive endocytic sites. We next sought to determine why the sites were nonproductive.

## WASP recruitment to endocytic sites depends on overlapping functions of SH3 and proline-rich domains of multivalent endocytic linker proteins

Previous in vitro studies showed that Sla1 interacts with the yeast WASP Las17 through SH3-PRM interactions (*Feliciano and Di Pietro, 2012*; *Rodal et al., 2003*). We therefore examined where Las17 is located relative to the nonproductive endocytic sites in *sla1 W41AW108A* and *pan1ΔPRD* single and double mutants.

In wild-type cells, Sla1 and Las17 accumulate and then leave endocytic sites with similar kinetics (*Figure 3A*). In *sla1W41AW108A* cells, Las17 still colocalizes with sla1W41AW108A patches (*Figure 3B*). However, Las17-GFP reaches its maximum fluorescence intensity approximately 20 s after sla1W41AW108A. More importantly, Las17-GFP fluorescence intensity reaches only half of its maximum value and continues to increase when sla1W41AW108A-mCherry fluorescence intensity has already begun to decline. Thus, in *sla1W41AW108A* cells, cortical recruitment of Las17 is no longer synchronized with sla1W41AW108A recruitment. These results are consistent with a role for Sla1 in Las17 recruitment (*Feliciano and Di Pietro, 2012*). However, Las17 is still recruited to endocytic sites in *sla1W41AW108A* cells, indicating that additional proteins help to recruit Las17. In *pan1ΔPRD* cells, Las17-GFP reaches its maximum fluorescence intensity about 5 s after Sla1-mCherry (*Figure 3C*). Thus, Las17 recruitment is also defective in *pan1ΔPRD* cells, although to a lesser extent than in *sla1W41AW108A* cells.

In *sla1W41AW108A pan1ΔPRD* cells, cortical sla1W41AW108A patches stay non-motile at the cell cortex, while Las17 patches move dynamically along the cell cortex and throughout the cytoplasm (*Figure 3D–F* and *Video 1*). Most cortical sla1W41AW108A patches did not recruit any detectable Las17 during a 2 min movie (95.4%, 131 patches from 14 cells were examined) (*Figure 3F*). In rare cases (4.6%), Las17 patches appeared to transiently colocalize with cortical sla1W41AW108A patches (*Figure 3F*). However, unlike in wild-type cells, neither sla1W41AW108A patches nor Las17 patches disassemble after the transient colocalization (*Figure 3F*). A likely explanation for such transient colocalization is that Las17 patches move along the cortex and move to or near the non-motile

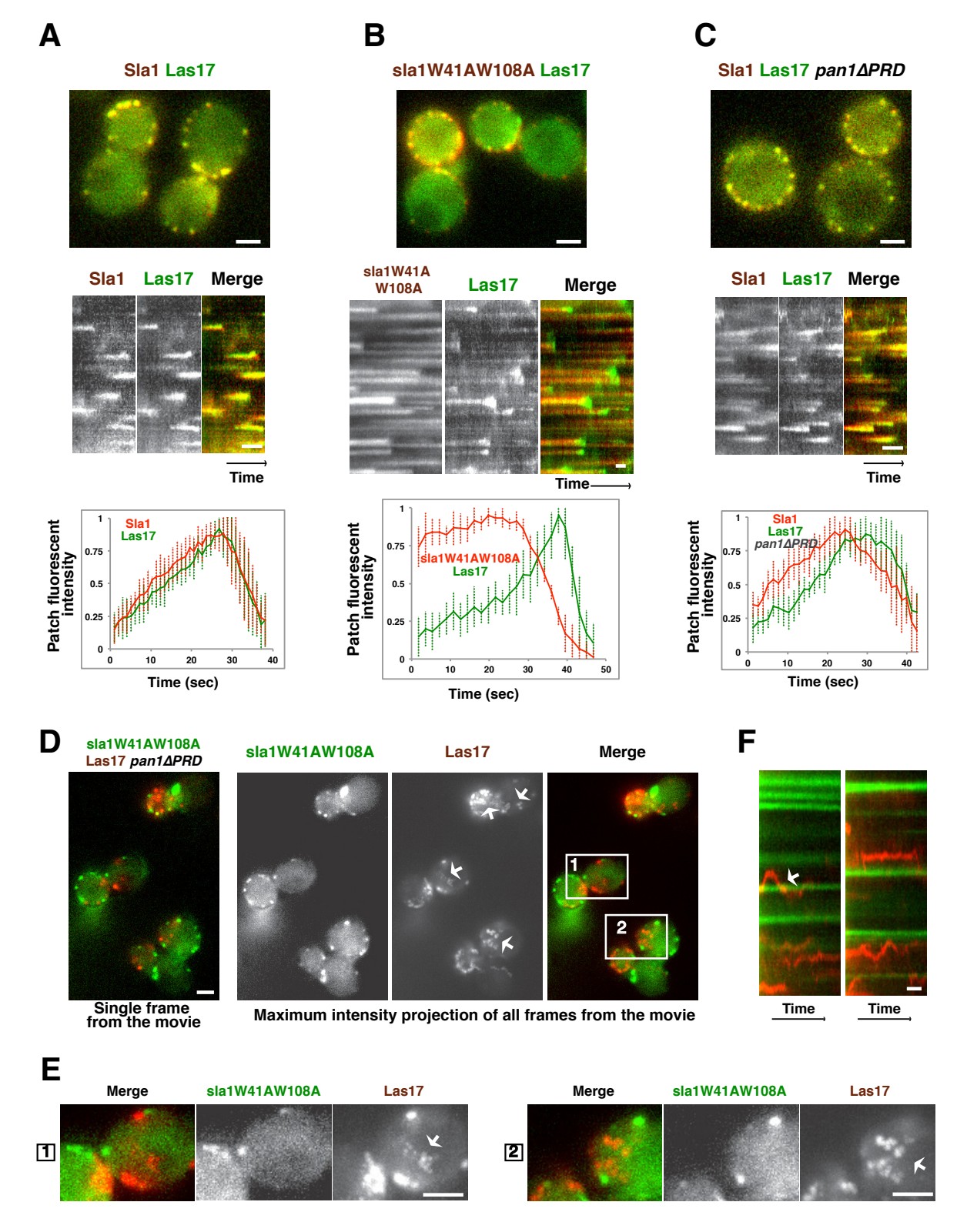

**Figure 3.** Las17 (yeast WASP) is not recruited to cortical endocytic sites in *sla1W41AW108A pan1ΔPRD* mutant cells. (**A-C**) Single frames (top) from movies and circumferential kymograph representations (middle) of GFP- and mCherry-tagged proteins. Averaged (mean ±SD) fluorescent intensity profiles for GFP- and mCherry-tagged proteins from 10 individual patches (bottom). (**A**) *SLA1-mCherry LAS17-GFP* cells. (**B**) *sla1W41AW108A-mCherry LAS17-GFP* cells. Note that the fluorescence intensity profile for this strain was only analyzed for the last 50 s of the patch lifetime. (**C**) *SLA1-mCherry*

*Figure 3 continued on next page*

*Figure 3 continued*

*LAS17-GFP pan1∆PRD* cells. (D and E) A single frame and maximum intensity projection of all frames from a movie (*Video 1*) of *sla1W41AW108A-GFP LAS17-TagRFP-T pan1∆PRD* cells. (E) Enlarged views of the boxed-areas shown in D. The arrows indicate Las17 patches in cytoplasm. (F) Circumferential kymograph representation of sla1W41AW108A-GFP and Las17-TagRFP-T in *pan1∆PRD*cells. The arrow indicates that a Las17-TagRFP-T patch transiently colocalizes with a static sla1W41AW108A-GFP patch. The scale bars on kymographs are 20 s. The scale bars on cell pictures are 2 µm.

sla1W41AW108A patches by chance. Thus, in s*la1W41AW108A pan1∆PRD* cells, Las17 is no longer recruited to cortical endocytic sites.

## Actin comet tails move throughout the cytoplasm when SH3 and proline-rich domains of multivalent endocytic linker proteins are absent

Since Las17 is not recruited to cortical endocytic sites in s*la1W41AW108A pan1∆PRD* cells, we asked how actin dynamics are affected. Fluorescently tagged-Abp1 (Actin binding protein 1) and Sac6 (yeast fimbrin), which both localize to cortical actin patches, were used to monitor endocytic actin assembly in the following experiments. These actin markers were used interchangeably because neither detectably altered actin function or dynamics.

In wild-type, *pan1∆PRD*, or *sla1W41AW108A* cells, the cortical actin patch marker Abp1(Actin binding protein 1) appears at endocytic sites at the end of the Sla1 (or sla1 mutant) lifetime, and both proteins are internalized and disappear (*Figures 4A, B and C*). Strikingly, instead of forming cortical patches in *sla1 W41AW108A pan1∆PRD* cells, actin formed comet tails, as seen with Abp1-RFP (*Figure 4D*) or Sac6-GFP (*Figure 4E*). Multifocus microscopy (MFM) (*Abrahamsson et al., 2013*) clearly demonstrated that these actin comet tails slide along the cell cortex or move through the cytoplasm over time (*Figure 4E*, *Video 2*). More importantly, these actin comet tails do not appear to originate at or associate stably with the cortical static sla1W41AW108A patches in *sla1 W41AW108A pan1∆PRD* cells (*Figure 4D*, *Video 3*). We examined 304 cortical static sla1-W41AW108A patches in 30 *sla1W41AW108A pan1∆PRD* cells and found that only 32 (10.5%) of them transiently colocalized with actin comet-tails during a 3 min movie (*Figure 4D*). However, unlike in wild-type cells, the sla1W41AW108A-GFP patches remained stationary at the cell cortex after these rare, transient colocalizations (*Figure 4D*). Thus, actin comet tails may move to or near the nonmotile sla1W41AW108A/ pan1∆PRD patches by chance when the comet tails move along the cell cortex.

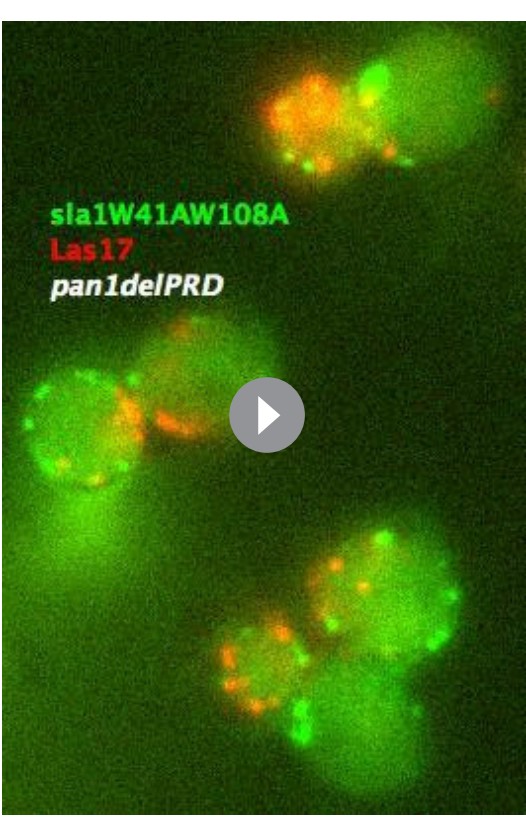

**Video 1.** Dynamics of sla1W41AW108A-GFP and Las17-TagRFP-T in *sla1W41AW108A-GFP LAS17-TagRFP-T pan1∆PRD* cells. Time to acquire one image pair is 1.8 s. Interval between frames is 1.8 s.

## WASP-Myosin module proteins localize at the leading tip of actin comet tails when SH3 and proline-rich domains of multivalent endocytic linker proteins are absent

As we showed above, Las17 patches and actin comet-tails are not associated with the nonproductive cortical endocytic sites in *sla1-W41AW108A pan1∆PRD* cells. We next determined the spatial relationship between Las17 and actin comet tails in this mutant.

Interestingly, in the s*la1W41AW108A pan1∆PRD* mutant, Las17 patches localized at

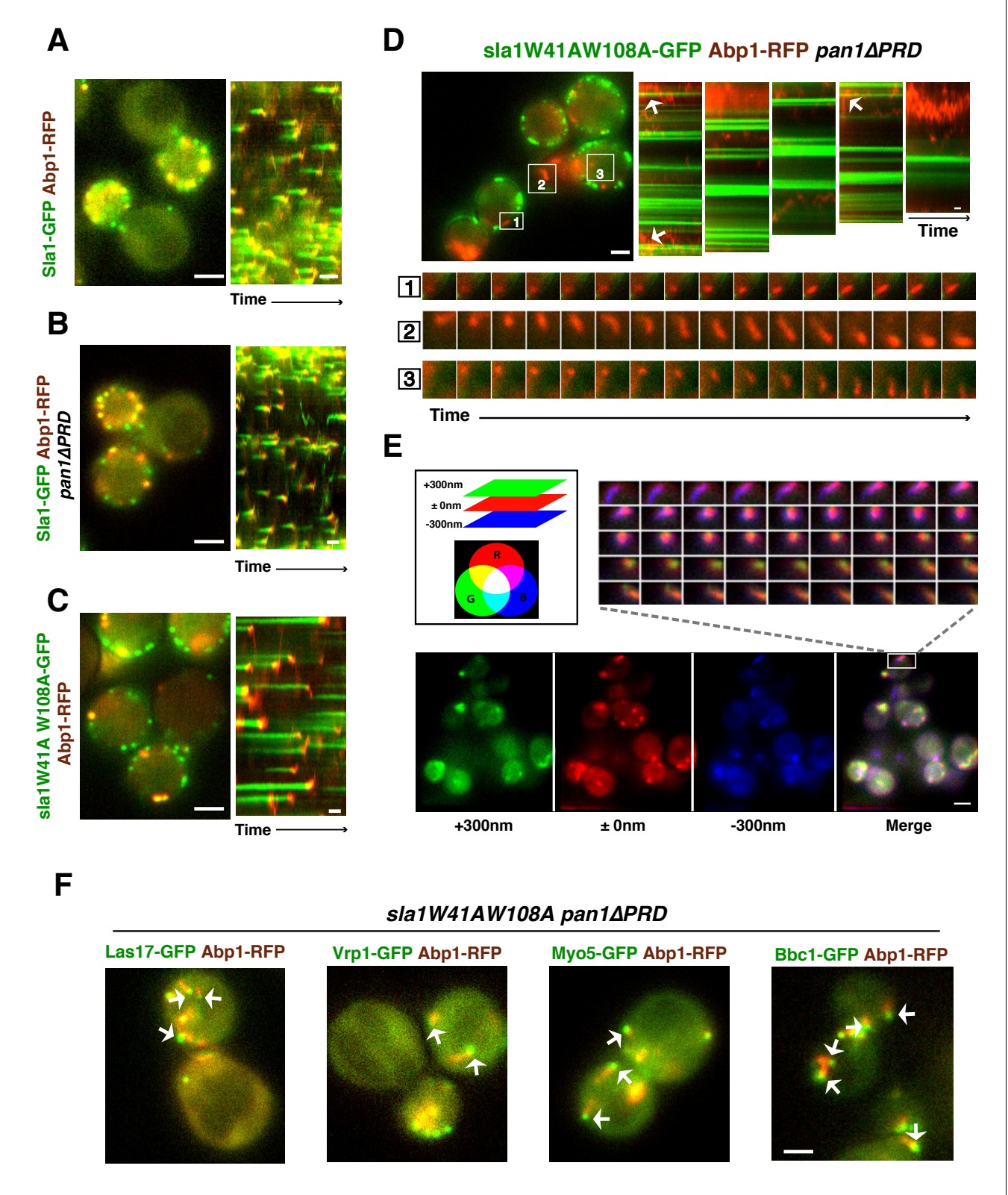

**Figure 4.** WASP-Myosin module proteins localize at the leading tip of actin comet tails in *sla1W41AW108A pan1ΔPRD* mutant cells. (A-C) Single frames (left) from movies and circumferential kymograph presentations (right) of GFP- and RFP-tagged proteins. (A) *SLA1-GFP ABP1-RFP* cells. (B) *SLA1-GFP ABP1-RFP pan1ΔPRD* cells. (C) *sla1W41AW108A-GFP ABP1-RFP* cells. (D) Single frame (top left) from movie (*Video 3*) and circumferential kymograph representations (top right) of sla1W41AW108A-GFP and Abp1-RFP in *pan1ΔPRD* cells. Montages of individual actin comet tails in the boxed areas in
*Figure 4 continued on next page*

*Figure 4 continued*

the top left image (bottom). (E) Dynamics of Sac6-GFP in *sla1W41AW108A pan1ΔPRD* cells observed by multifocus microscopy (MFM). An example of three simultaneously acquired Z-planes that are artificially colored in green, red or blue (bottom panel). Note that the actin tails in the merged image reveal different colors depending on their positions in Z-planes. Montage of one actin comet tail from a movie (*Video 2*) acquired in multiple Z-planes simultaneously at 1 frame/250 msec (top right). Note that the actin tail changes colors over time, reflecting movement through the cytoplasm. (F) Single frames from movies (*Videos 4* and *5*) of cells expressing indicated GFP-tagged protein and Abp1-RFP. The arrows indicate that WASP-Myosin module proteins localize at the leading tip of actin comet tails in s*la1W41AW108A pan1ΔPRD* mutants. The scale bars for kymographs are 20 s. The scale bars for cell pictures are 2 μm.

The following figure supplement is available for figure 4:

**Figure supplement 1.** Localization of Ede1 in *sla1W41AW108A pan1ΔPRD* mutant cells.

the leading tips of actin comet tails and are apparently propelled by actin assembly in a manner similar to actin rocket tails on pathogens such as *vaccinia virus* or *Shigella flexneri* (*Welch and Way, 2013*) (*Figure 4F*, *Video 4*). Similar to Las17, several other components of WASP-Myosin module, including Vrp1, Myo5, and Bbc1, were also observed at the leading tip of actin comet tails in *sla1-W41AW108A pan1ΔPRD* cells (*Figure 4F*, *Video 4*, and *Video 5*). These results show that the actin comet tails share molecular characteristics with the endocytic actin machinery. Thus, the endocytic actin machinery no longer assembles at cortical endocytic sites in *sla1W41AW108A pan1ΔPRD* cells, but at distinct sites, the nature of which is presently obscure. Consistently, the yeast Eps15, Ede1, which is one of the early module proteins (*Figure 1A*) and functions in endocytic site initiation and stabilization (*Kaksonen et al., 2005*), is not detected at the tip of the actin comet tails (*Figure 4—figure supplement 1*). These results demonstrate that the WASP-Myosin module proteins can induce actin assembly and forces independent of the upstream endocytic machinery.

Together, our results indicate that two Sla1 SH3 domains and the Pan1 PRD together play indispensable roles in coupling the WASP-Myosin machinery to endocytic sites. Thus, we conclude that WASP-Myosin module proteins are recruited to endocytic sites mainly through SH3-PRM interactions (Sla1 recruits PRD-containing proteins and Pan1 recruits SH3 domain-containing proteins). Previous in vitro results suggested that Sla1 SH3 domains inhibit Las17 NPF activity (*Feliciano and Di Pietro, 2012*; *Rodal et al., 2003*), while the Pan1 PRD activates Myo3/5 NPF activity (*Barker et al., 2007*). Next, we addressed the recruitment and NPF regulatory roles of the Sla1 SH3 domains and the Pan1 PRD in WASP-Myosin module regulation.

## Restoring WASP or WIP cortical localization by end3C fusion compensates for loss of interactions mediated by multivalent endocytic linker proteins

The *sla1W41AW108A pan1ΔPRD* mutant provides a unique opportunity to identify the key player(s) that recruit the actin assembly machinery to endocytic sites and to explore which

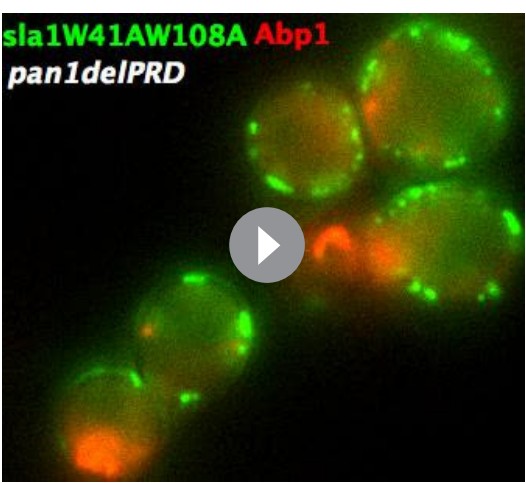

**Video 3.** Dynamics of sla1W41AW108A-GFP and Abp1-RFP in *sla1W41AW108A-GFP ABP1-RFP pan1ΔPRD* cells. Time to acquire one image pair is 1.8 s. Interval between frames is 1.8 s.

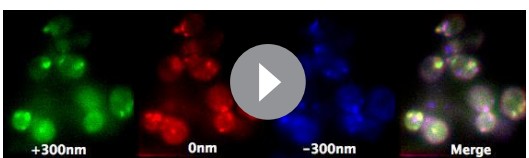

**Video 2.** Dynamics of Sac6-GFP in *sla1W41AW108A pan1ΔPRD* cells captured by multifocus microscopy (MFM). Time interval between frames is 250 ms.

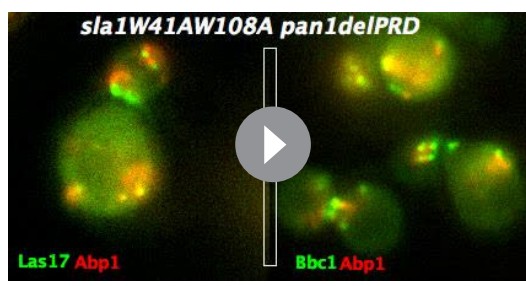

**Video 4.** Dynamics of Las17-GFP or Bbc1-GFP and Abp1-RFP in *sla1W41AW108A pan1ΔPRD* cells. Time to acquire one image pair is 1.8 s. Interval between frames is 1.8 s.

parameters are important for triggering actin assembly. Thus, we developed a strategy to artificially direct selected proteins to cortical sla1-W41AW108A/pan1ΔPRD sites, and then tested whether cell growth and productive endocytic actin assembly were restored.

Previous studies reported that End3's C-terminus (end3C) interacts with the Pan1 central region with high affinity ($K_d$ = 27 nM) (*Boeke et al., 2014*; *Sun et al., 2015*). This C-terminal region contains less than 200 amino acids (*Figure 1—figure supplement 1A*) and its primary function is to mediate cortical recruitment through interaction with the Pan1 N-terminus (*Sun et al., 2015*; *Tang et al., 2000*). These features make end3C an appealing candidate to recruit Las17 to pan1ΔPRD sites in *sla1-W41AW108A pan1ΔPRD* double mutants. We generated a *LAS17-end3C* strain and tagged Las17-end3C with GFP. The sequence encoding Las17-end3C-GFP was integrated into the *LAS17* chromosomal locus so the hybrid gene was expressed from *LAS17*'s endogenous promoter. *las17WCAΔ myo5CAΔ myo3Δ* cells exhibit severe growth defects due to the loss of NPF activity from both NPFs (*Sun et al., 2006*). However, *LAS17-end3C-GFP myo5CAΔ myo3Δ* cells grow well (*Figure 5—figure supplement 1A*). We conclude that the end3C fusion does not interfere Las17's NPF activity.

Strikingly, Las17-end3C-GFP restored *sla1W41AW108A pan1ΔPRD* cells to normal growth at not only 25°C and 30°C, but also the non-permissive temperature of 37°C (*Figure 5A*). Remarkably, Las17-end3C-GFP even rescued *sla1Δ pan1ΔPRD* from lethality at all temperatures (*Figure 5A* and *Figure 5—figure supplement 1B*). Las17-end3C-GFP in *LAS17-end3C-GFP sla1W41AW108A pan1ΔPRD* cells is expressed at levels indistinguishable from Las17-GFP in wild-type cells (*Figure 5—figure supplement 1C*). Furthermore, there is no significant difference in patch maximum fluorescence intensity between Las17-end3C-GFP in the *LAS17-end3C-GFP sla1W41AW108A pan1ΔPRD* mutant and Las17-GFP in wild-type cells (*Figure 5—figure supplement 1D and E*). In addition, Las17-end3C-GFP patches only appear at the cell cortex and they develop fluorescence intensity with similar kinetics to pan1ΔPRD-mCherry in the *sla1W41AW108A pan1ΔPRD* mutant (*Figure 5—figure supplement 1F*), indicating that end3C is sufficient to recruit Las17 to cortical endocytic sites through its interaction with pan1ΔPRD when the Sla1 SH3- and Pan1 PRD- mediated interactions are absent. We conclude that artificial Las17 recruitment bypasses the requirement for the Sla1 SH3 domains and Pan1 PRD for normal cell growth.

We next fused end3C to the C-terminus of additional SH3 domain- or PRD-containing endocytic proteins (*Figure 5—figure supplement 2*) and determined whether they could also restore normal growth to *sla1W41AW108A pan1ΔPRD* cells (*Figure 5B*). Intriguingly, Vrp1-end3C fully restored cell growth, similar to Las17-end3C. Bzz1-end3C (yeast functional homolog of TOCA-1, FBP17, CIP4, or PACSIN) and Rvs167-end3C (yeast Amphiphysin) also rescued cell growth, but to lesser extents, particularly at 37°C. In addition, Myo5-end3C rescued *sla1W41AW108A pan1ΔPRD* cell growth, but to a lesser extent than Bzz1-end3C or Rvs167-end3C. In contrast, Lsb4-end3C (yeast SH3YL1a) (*Figure 5B*) and Bbc1-end3C (data not shown) did not rescue cell growth. Thus, we have identified five PRD- or SH3 domain- containing endocytic proteins that can rescue cell growth of the *sla1W41AW108A pan1ΔPRD* mutant to different extents when they are fused to end3C.

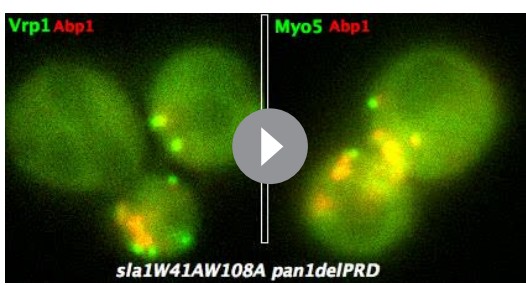

**Video 5.** Dynamics of Vrp1-GFP or Myo5-GFP and Abp1-RFP in *sla1W41AW108A pan1ΔPRD* cells. Time to acquire one image pair is 1.8 s. Interval between frames is 1.8 s.

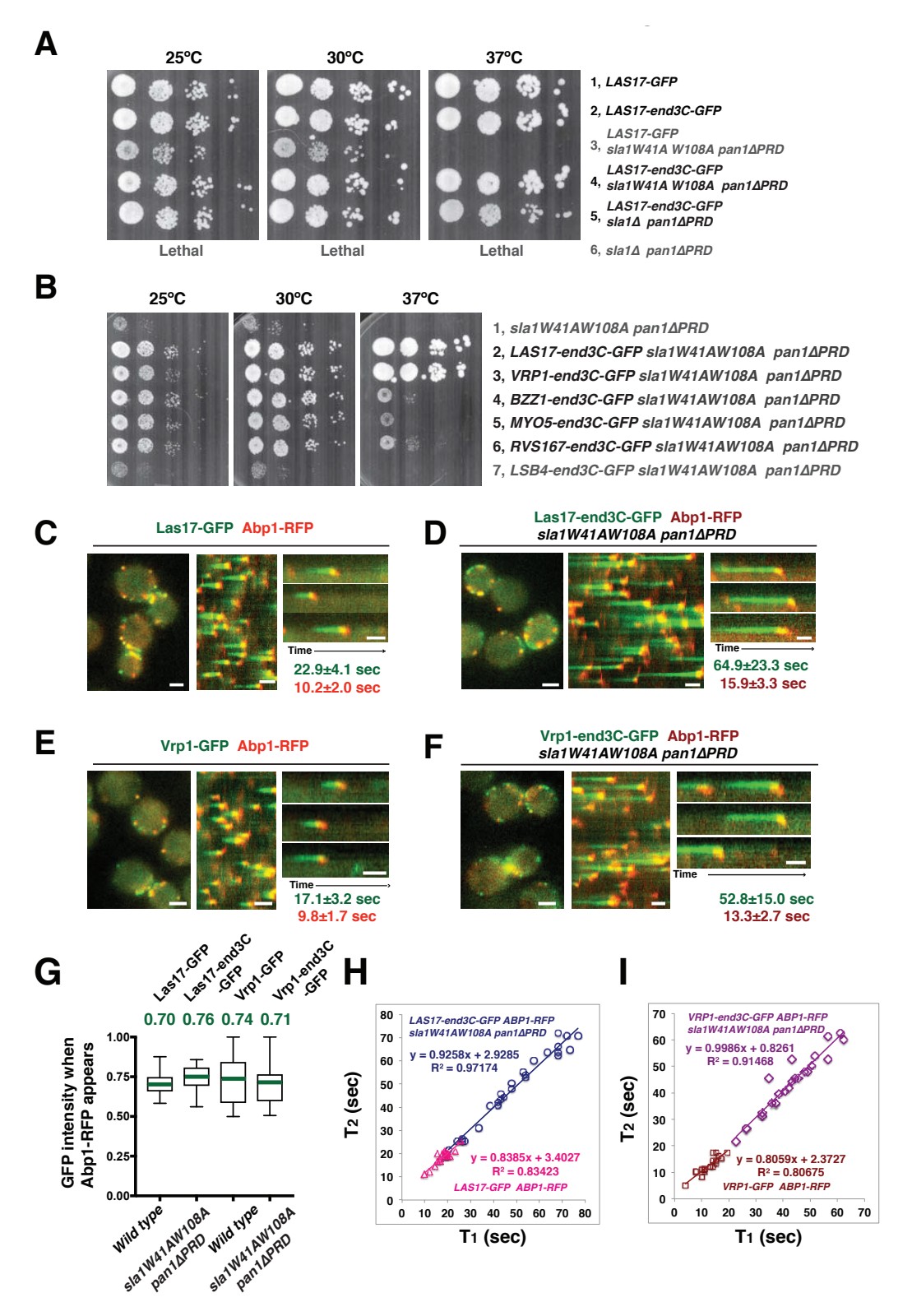

**Figure 5.** A WASP-end3C chimeric protein or a WIP-end3C chimeric protein restores normal growth and productive endocytic actin assembly in an *sla1W41AW108A pan1ΔPRD* mutant. The onset of actin assembly is tightly coupled to accumulation of a threshold of WASP or WIP at endocytic sites. (**A** and **B**) Cell growth at 25°C, 30°C or 37°C of indicated yeast strains spotted as serial dilutions of liquid cultures on plates (also see *Figure 5—figure supplement 1B*). (**C** and **E**) Single frame (left) from movie (*Video 6* or *Video 7*), circumferential kymograph representation (middle), and radial

*Figure 5 continued on next page*

Figure 5 continued

kymograph representations (right) of Las17-GFP and Abp1-RFP, or Vrp1-GFP and Abp1-RFP in wild-type cells. (D and F) Single frame (left) from movie (*Video 6* and *Video 7*), circumferential kymograph presentation (middle), and radial kymograph representation (right) of Las17-end3C-GFP and Abp1-RFP, or Vrp1-end3C-GFP and Abp1-RFP in *sla1W41AW108A pan1ΔPRD* cells. The numbers shown in C-F are lifetime of GFP-tagged protein (in green) and Abp1-RFP (in red) in the indicated strains (also see *Figure 5—figure supplement 3A*). (G) The average fluorescence intensity of GFP-tagged patch proteins at the moment when the Abp1-RFP signal appear at endocytic sites for the indicated strains. H and I, $T_1$vs $T_2$ plots for indicated strains (for the details, please see *Figure 5—figure supplement 3B*). The scale bars in kymographs are 20 s. The scale bars on cell pictures are 2 μm.

The following figure supplements are available for figure 5:

**Figure supplement 1.** The end3C fusion does not interfere with Las17's NPF activity and is sufficient to recruit Las17 to cortical endocytic sites in *sla1W41AW108A pan1ΔPRD* mutant cells.

**Figure supplement 2.** Protein expression and dynamics of various end3C fused proteins.

**Figure supplement 3.** Quantification of endocytic patch protein lifetime for strains shown in *Figure 5C–F* and flowchart for scheme used to plot graphs shown in *Figure 5H and I*.

Since end3C fused to Las17 (yeast N-WASP) or Vrp1 (yeast WIP) restored *sla1W41AW108A pan1ΔPRD* to normal growth (*Figure 5A and B*), we examined endocytic actin assembly in these two strains. As in wild-type cells (*Figure 5C and E*), Abp1-RFP labeled cortical actin patches moved inward and then disappeared in both *LAS17-end3C-GFP sla1W41AW108A pan1ΔPRD* (*Figure 5D*) (*Video 6*) and *VRP1-end3C-GFP sla1W41AW108A pan1ΔPRD* cells (*Figure 5F*) (*Video 7*), indicating that the actin assembly is productive and that endocytic internalization is restored in the mutant strains. Consistently, cytoplasmic actin comet tails were no longer observed in these two mutants (*Figure 5D and F*, *Videos 6* and *7*). The patch lifetimes of Las17-end3C-GFP or Vrp1-end3C-GFP in *sla1W41AW108A pan1ΔPRD* cells, respectively, were less regular and approximately two times longer than the lifetimes of Las17 or Vrp1 in wild-type cells (*Figure 5C–F* and *Figure 5—figure supplement 3A*). In contrast, the actin patch (Abp1-RFP) lifetimes were much less affected in *LAS17-end3C-GFP sla1W41AW108A pan1ΔPRD* or *VRP1-end3C-GFP sla1W41AW108A pan1ΔPRD* cells compared to wild-type cells. These results demonstrate that restoring Las17 or Vrp1 cortical localization using artificial, engineered fusions, is sufficient to direct productive actin assembly to cortical endocytic sites to compensate for loss of interactions mediated by Sla1 SH3 domains and the Pan1 PRD. Thus, our data indicate that Sla1 SH3 domains and the Pan1 PRD primarily function in recruitment, rather than the NPF regulation, of the WASP-Myosin module.

## WASP NPF activity at endocytic sites does not appear to be regulated in the previously assumed manner

Previous studies suggest that in contrast to its mammalian N-WASP homologue, Las17's actin nucleation promoting activity is not autoinhibited in vitro (*Feliciano and Di Pietro, 2012*; *Rodal et al., 2003*). However, actin assembly is only observed approximately 15 s after Las17 is recruited to endocytic sites in wild-type cells (*Figure 5C*). The interaction between the first two Sla1 SH3 domains and the Las17 PRD is thought to be important for keeping Las17 inactive until actin assembly starts (*Feliciano and Di Pietro, 2012*; *Rodal et al., 2003*). In *sla1W41AW108A pan1ΔPRD* cells, the Las17-end3C nucleation promoting activity can no longer be inhibited by sla1W41AW108A due to the lack of functional Sla1 SH3 domains, so actin assembly might occur prematurely. However, in the *sla1-W41AW108A pan1ΔPRD* mutant, Las17-end3C-GFP persisted at endocytic sites even longer than in wild-type cells before actin assembly was

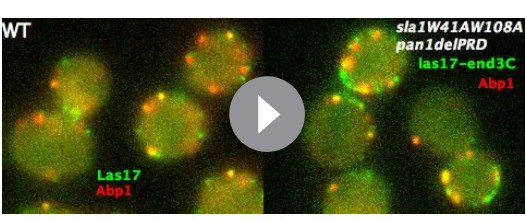

**Video 6.** , Dynamics of Las17-GFP and Abp1-RFP in wild type cells and dynamics of Las17-end3C-GFP and Abp1-RFP in *sla1W41AW108A pan1ΔPRD* cells. Time to acquire one image pair is 1.1 s. Interval between frames is 1.1 s.

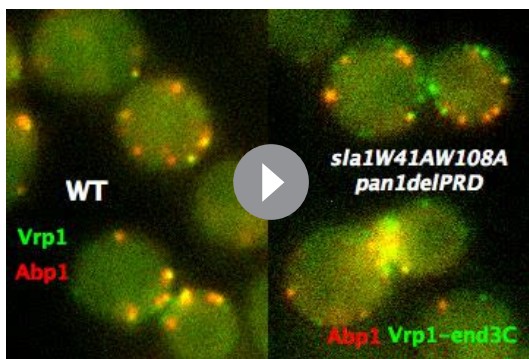

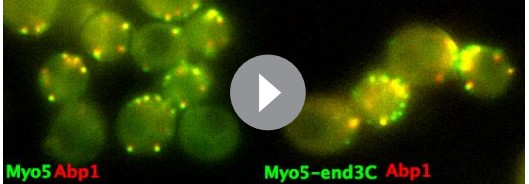

**Video 8.** Dynamics of Myo5-GFP and Abp1-RFP in wild-type cells and dynamics of Myo5-end3C-GFP and Abp1-RFP in *myo3Δ* cells. Time to acquire one image pair is 1.0 s. Interval between frames is 1.0 s.

**Video 7.** , Dynamics of Vrp1-GFP and Abp1-RFP in wild-type cells and dynamics of Vrp1-end3C-GFP and Abp1-RFP in *sla1W41AW108A pan1ΔPRD* cells. Time to acquire one image pair is 1.2 s. Interval between frames is 1.2 s.

initiated (*Figure 5D*). These results indicate that Las17 NPF activity at endocytic sites does not appear to be regulated as was previously assumed. We next investigated which parameters are important for the onset of actin assembly.

## The onset of productive actin assembly at endocytic sites appears tightly coupled to accumulation of WASP and WIP to threshold levels

We asked whether the quantity of Las17 or Vrp1 at endocytic sites predicts endocytic actin assembly initiation. The fluorescence intensity of the GFP-tagged proteins at the moment when the Abp1-RFP signal appeared (the onset of actin assembly) at endocytic sites was determined for the strains shown in *Figure 5C–F*. Interestingly, regardless of differences in lifetimes, the average intensities of the GFP-tagged Las17 and Vrp1 were all in a similar range (70–80% of their maximum intensity) when actin assembly was initiated (*Figure 5G* and *Figure 5—figure supplement 3B*). Importantly, the end3C-fusions did not alter expression levels of the tagged proteins (*Figure 5—figure supplement 1C* and *Figure 5—figure supplement 2A*). Thus, these results establish that productive actin assembly is initiated when similar numbers of Las17 and Vrp1 are recruited to endocytic sites in wild-type and *LAS17-end3C-GFP sla1W41AW108A pan1ΔPRD* cells. Furthermore, when the time, T1, during at which the GFP-tagged protein reaches its average intensity, was plotted against the time T2, when actin assembly is first detected, for numerous individual endocytic events in these strains (*Figure 5H–I*, *Figure 5—figure supplement 3B*), the high R-squared values of the linear trendlines drawn for these plots indicated that a threshold accumulation (70–80% of their maximum quantity at endocytic sites) of Las17 or Vrp1 at endocytic sites is tightly correlated with the onset of productive actin assembly in both wild-type and in *sla1W41AW108A pan1ΔPRD* cells.

## Productive endocytic actin assembly appears to occur in a switch-like manner upon WASP-WIP recruitment to a threshold level, regardless of how recruitment is induced

When Las17-end3C or Vrp1-end3C is recruited to endocytic sites through interaction of its end3C domain with pan1ΔPRD, other proteins associated with the cytoplasmic actin comet tails observed in *sla1W41AW108A pan1ΔPRD* cells presumably reach cortical endocytic sites by (direct or indirect) interactions with the end3C-fused protein.

We next quantitatively addressed how Las17-end3C or Vrp1-end3C recruit their binding partners to endocytic sites and facilitate productive actin assembly in *sla1W41AW108A pan1ΔPRD* cells.

In *sla1W41AW108A pan1ΔPRD* cells, Las17-end3C-GFP and Vrp1-mCherry, or Vrp1-end3C-GFP and Las17-TagRFP-T, respectively, developed fluorescence intensity with identical kinetics at cortical patches (*Figure 6A* and *Figure 6—figure supplement 1A and B*). Thus, when cortical localization of either Las17 or Vrp1 is restored through an end3C fusion in *sla1W41AW108A pan1ΔPRD* cells, the other protein is recruited simultaneously. This result explains why Las17-end3C and Vrp1-end3C rescue the *sla1W41AW108A pan1ΔPRD* mutant phenotypes to the same extent (*Figure 5B*). Moreover,

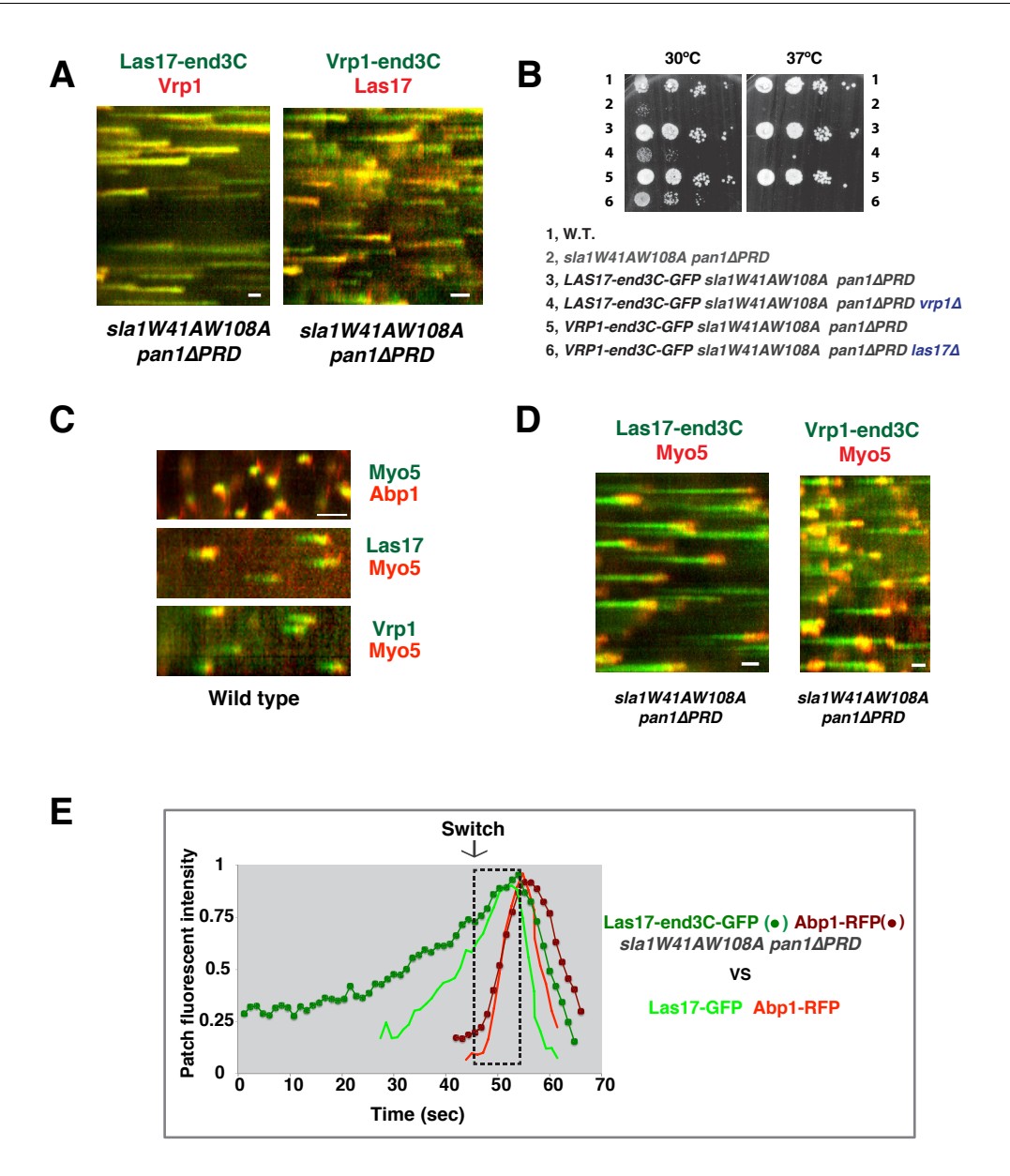

**Figure 6.** Actin assembly is triggered in a switch-like manner that corresponds to a threshold level of WASP-WIP accumulation. (**A**) Circumferential kymograph presentations of GFP- and mCherry- or TagRFP-T-tagged proteins for indicated yeast strains (also see *Figure 6—figure supplement 1*). (**B**) Examining growth of indicated yeast strains by spotting serial dilutions of liquid cultures on plate. (**C**) Circumferential kymograph presentations of GFP- and RFP- or TagRFP-T-tagged proteins in wild-type cells. (**D**) Circumferential kymograph presentations of GFP- and TagRFP-T-tagged proteins for indicated yeast strains. (**E**) Alignment of averaged intensity measurements of GFP- and RFP-tagged proteins from the indicated yeast cells. For more details, please also see *Figure 6—figure supplement 2*. The scale bars in kymographs are 20 s.

The following figure supplements are available for figure 6:

**Figure supplement 1.** Two-color fluorescence intensity profiles for Las17-end3C-GFP and Vrp1-mCherry, or Vrp1-end3C-GFP and Las17-TagRFP-T in *sla1W41AW108A pan1ΔPRD* mutant cells.

**Figure supplement 2.** Alignment of averaged (mean ±SD) intensity measurements for GFP- and RFP-tagged proteins in the indicated yeast strains (note different time scales in A and B).

our genetic studies indicated that Vrp1 or Las17 is required for Las17-end3C or Vrp1-end3C, respectively, to restore normal growth of *sla1W41AW108A pan1ΔPRD* cells (*Figure 6B*). On the other hand, similar to the wild-type cells (*Figure 5C and E*, *Figure 6C*), the onset of actin assembly appears to coincide with the recruitment of Myo5-GFP (type I Myosin) in *LAS17-end3C-GFP sla1-W41AW108A pan1ΔPRD* cells (*Figure 5D* and *Figure 6D*) and in *VRP1-end3C-GFP sla1W41AW108A pan1ΔPRD* cells (*Figure 5F* and *Figure 6D*). Taken together, Las17 and Vrp1 (recruited by end3C fusions) are necessary and sufficient to recruit the WASP-Myosin module proteins to endocytic sites and restore productive endocytic actin assembly, compensating for loss of interactions mediated by multivalent endocytic linker proteins.

To further assess the onset of actin assembly kinetically, we analyzed fluorescence intensity development of GFP- and RFP-tagged proteins in *LAS17-end3C-GFP ABP1-RFP sla1W41AW108A pan1ΔPRD* and *LAS17-GFP ABP1-RFP* cells (*Figure 5C and D*). As shown in *Figure 6E* and *Figure 6—figure supplement 2*, Abp1-RFP joins Las17-GFP or Las17-end3C-GFP patches when the GFP-tagged proteins reach 70–80% of maximum intensity in *LAS17-GFP ABP1-RFP* or *LAS17-end3C-GFP ABP1-RFP sla1W41AW108A pan1ΔPRD* cells. It takes much longer for Las17-end3C-GFP to reach 70–80% of its maximum intensity in *sla1W41AW108A pan1ΔPRD* cells than for Las17-GFP to reach those levels in wild-type cells. This delay reflects the much longer lifetime of Las17-end3C-GFP patches observed in *sla1W41AW108A pan1ΔPRD* cells (*Figure 5C and D*, *Figure 5—figure supplement 3A*). Thus, even though Las17 NPF activity cannot be inhibited by sla1W41AW108A, Las17-end3C-GFP does not appear to initiate actin nucleation until it reaches a 'threshold' of 70–80% of its maximum intensity. Intriguingly, once Las17-GFP or Las17-end3C-GFP reaches an apparent 'threshold' level, the Abp1-RFP signal increases rapidly and reaches its maximum level at a similar rate in both wild-type and *LAS17-end3C-GFP sla1W41AW108A pan1ΔPRD* cells (*Figure 6E*). Thus, productive endocytic actin assembly appears to be triggered in an 'all or nothing' manner once the quantity of Las17 (as well as Vrp1, based on *Figure 6—figure supplement 1A*) reaches a threshold level, regardless how recruitment occurs.

## Recruitment and NPF activation of type I myosin by WIP appears to occur in a switch-like manner

Previous studies indicated that Las17 and the Myo3/5 (type I myosin)-Vrp1 complex are the two major NPFs for endocytic actin assembly (*Sirotkin et al., 2005*; *Sun et al., 2006*). Vrp1 is required for Myo3/5's recruitment and NPF activation. As we showed above, Myo3/5 arrives at endocytic sites with similar timing to the onset of actin assembly in both wild-type and *sla1W41AW108A pan1ΔPRD* mutant cells (*Figure 5C–F*, *Figure 6C and D*). Thus, similar to actin assembly, Myo3/5 recruitment also appears to occur in a switch-like manner upon Vrp1 accumulation to a threshold level. To further explore this mechanism, we next examined how type I myosin recruitment affects the onset endocytic actin assembly. To do this, we altered the mechanism for type I myosin recruitment to endocytic sites by fusing end3C to Myo5 in *myo3Δ* strain. *MYO5-end3C las17WCAΔ myo3Δ* cells grow much better than *myo5CAΔ las17WCAΔ myo3Δ* cells, suggesting that NPF activity is retained in Myo5-end3C fusion protein (*Figure 5—figure supplement 2D*).

As expected, Myo5-end3C-GFP was now recruited (through an end3C-Pan1 interaction) to endocytic sites much earlier than in wild-type cells relative to other proteins such as Las17 (*Figure 6C* and *Figure 7A*), and it had a longer lifetime (*Figure 7B*). The timing of recruitment and lifetime of Myo5-end3C-GFP are similar to those of Las17 and Vrp1 (*Figure 7A and B*)(*Sun et al., 2006*). However, actin assembly was not triggered during the first 12.6 ± 4.9 s of Myo5-end3C-GFP lifetime (*Figure 7A–C*, *Video 8*), even though Myo5-end3C and Vrp1 (as well as Las17) were both present at cortical patches (*Figure 7A*). Thus, the co-existence of type I Myosin and Vrp1 at endocytic sites is not sufficient to trigger actin assembly. Strikingly, in these mutant cells in which type I myosin arrives at endocytic sites with altered timing, the onset of actin assembly still coincided with the same apparent threshold level of Vrp1 recruitment (*Figure 7D* and *Figure 7—figure supplement 1*). These results suggest that NPF activation of type I myosin by Vrp1 also occurs in an 'all or nothing' manner. More importantly, these data further support a model in which recruitment of a threshold level of Las17 and Vrp1 plays a decisive role in switch-like initiation of productive actin assembly in vivo.

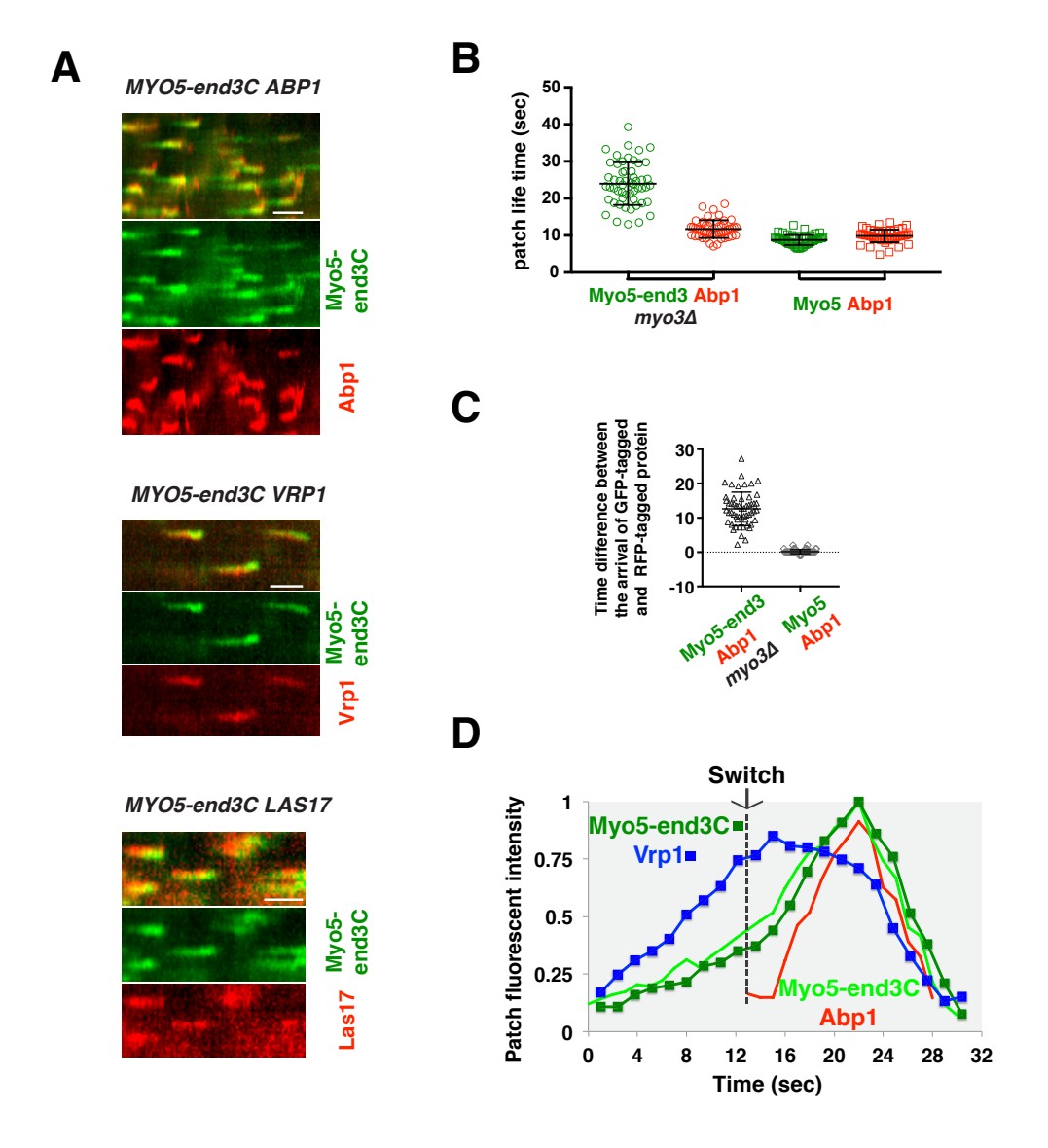

**Figure 7.** The onset of actin assembly coincides with WIP recruitment to a threshold level in cells expressing Myo5-end3C-GFP. (**A**) Circumferential kymograph presentations of GFP- and RFP- or TagRFP-T-tagged proteins in the indicated yeast strains. The *MYO3* gene was knocked out in all strains. (**B**) Lifetimes of GFP- and RFP-tagged proteins in the indicated strains. (**C**) Time difference between the arrival of GFP- and RFP tagged proteins in the indicated strains. (**D**) Alignment of averaged intensity profiles of GFP- and mCherry or RFP-tagged proteins in the indicated yeast cells. For more details, please also see *Figure 7—figure supplement 1*. The scale bars in kymographs are 20 s.

The following figure supplement is available for figure 7:

**Figure supplement 1.** Alignment of averaged (mean ±SD) intensity measurements for GFP- and mCherry or RFP-tagged proteins from the indicated yeast strains.

## Discussion

A core actin force-generating machine consisting of the Arp2/3 complex, NPFs and type 1 myosin generates forces on membranes for a wide variety of biological processes, from yeast to humans (*Cheng et al., 2012*; *Gupta et al., 2013*; *Kim and Flavell, 2008*; *Krendel et al., 2007*; *Lewellyn et al., 2015*; *McIntosh and Ostap, 2016*; *Sirotkin et al., 2005*; *Sun et al., 2006*). How this machine is recruited to a specific membrane domain and the principles governing actin assembly

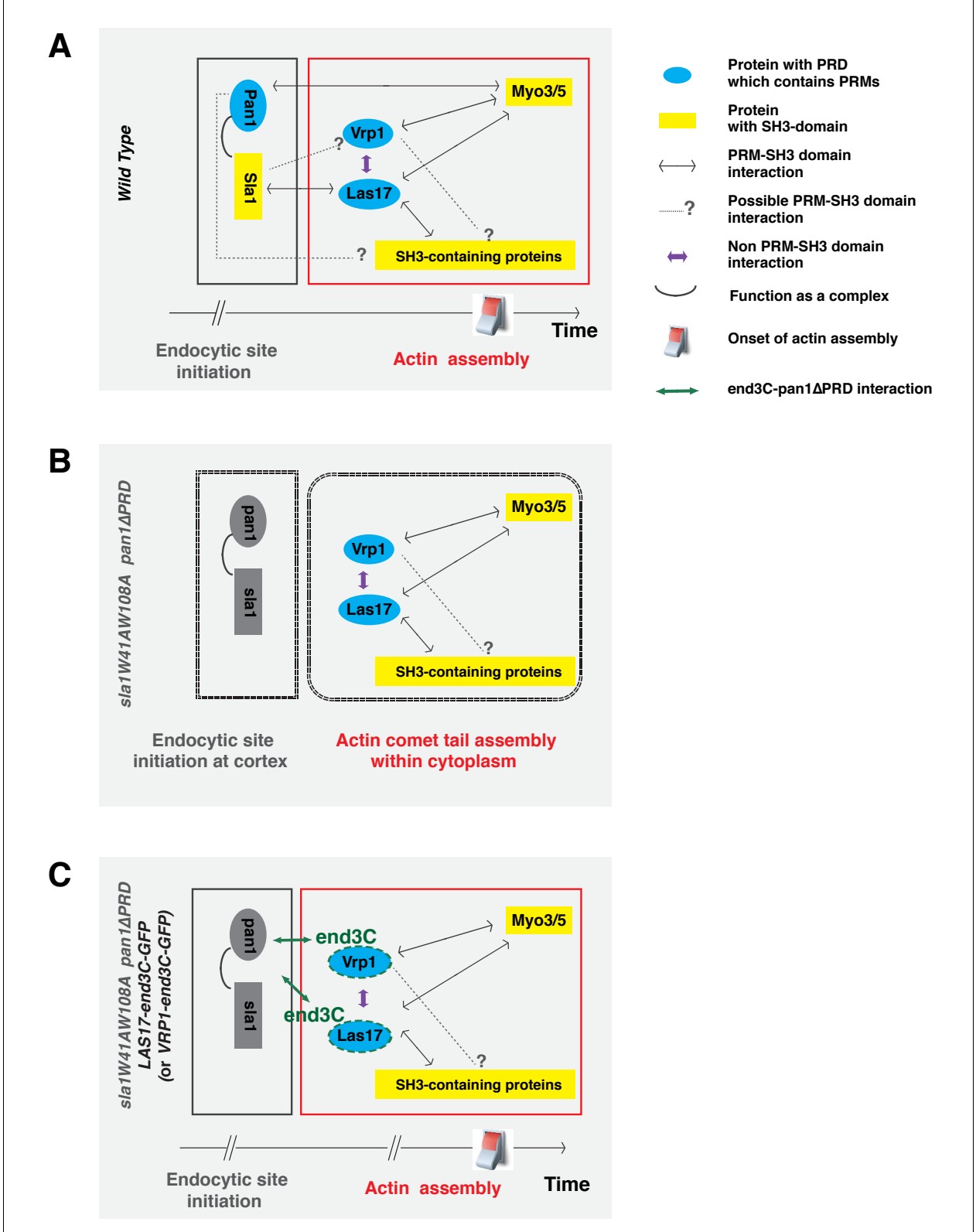

**Figure 8.** Switch-like activation of the Arp2/3 complex mediated by SH3 domain-PRM mediated interactions during yeast CME. Models for roles of SH3 domain-PRM interactions in spatiotemporal regulation of actin assembly in wild-type cells (**A**), in *sla1W41AW108A pan1ΔPRD* cells (**B**), in *sla1W41AW108A pan1ΔPRD LAS17-end3C-GFP* or *sla1W41AW108A pan1ΔPRD VRP1-end3C-GFP* cells (**C**). Blue represents PRD containing proteins. Yellow represents SH3 domain-containing proteins. The gray two-way arrows indicate SH3 domain-PRM interactions previously demonstrated by in vitro

*Figure 8 continued on next page*

*Figure 8 continued*

assays. The dotted line with a question mark indicates possible SH3 domain-PRM interactions. The purple arrows indicate non-SH3 domain-PRM interactions that have previously been identified. The green arrow indicates the interaction between end3C and pan1ΔPRD. The grey half circle indicates that multiple proteins function as a complex. The switch symbol marks the onset of actin assembly. See Discussion for description.

and force generation are still not well understood in vivo due to factors including the complexity of the network of protein-protein interactions involved (*Johnson and Hummer, 2013*; *Takenawa and Suetsugu, 2007*) and the apparent functional redundancy (*Galletta et al., 2008*; *Lewellyn et al., 2015*; *Sirotkin et al., 2005*; *Sun et al., 2006*). Here, we used genetics and live-cell imaging to investigate the molecular principles governing the recruitment and activation of this actin assembly and force-generating machine in vivo.

## SH3 domain-PRM interactions involving the yeast counterparts of intersectin couple the actin machinery to endocytic sites

The yeast WASP Las17 PRD and the type I myosin Myo3/5 SH3 domain have previously been shown to interact with Sla1's SH3 domains and with Pan1's PRD in vitro, respectively, to regulate Las17 and Myo3/5-Vrp1 NPF activity (*Barker et al., 2007*; *Feliciano and Di Pietro, 2012*; *Rodal et al., 2003*). However, the results from the current study strongly indicate that the crucial in vivo function of two Sla1 SH3 domains and the Pan1 PRD is instead to recruit components of the WASP-Myosin module through formation of a robust interaction network (discussed further in the following sections), triggering actin assembly and orchestrating force production at endocytic sites.

We propose that the Sla1 SH3 domains play a primary role in guiding WASP-Myosin module proteins to endocytic sites by interacting with Las17 (which in turn interacts with Vrp1), while the Pan1 PRD plays a supportive role (*Figure 8A*). In mutant cells in which either the two Sla1 SH3 domains are mutated (*Figure 3B*) or the Las17 PRD is partially truncated (*Feliciano and Di Pietro, 2012*), coordination of cortical Las17 and Sla1 recruitment is disrupted. In these cases, the Pan1 PRD can still interact with Myo3/5 (*Barker et al., 2007*), which interacts with Vrp1 and Las17 (*Evangelista et al., 2000*), to organize the actin machinery at endocytic sites (*Figure 8A*). It is also possible that the Pan1 PRD interacts with other SH3 domain-containing proteins, which interact with Las17. When both the Sla1 SH3 domains and the Pan1 PRD are mutated, the WASP-Myosin module components fail to be recruited to endocytic sites (*Figure 8B*). Future studies should explore whether the Pan1 PRD and/or Vrp1 PRD interact with other SH3 domain-containing endocytic proteins besides Myo3/5 (*Figure 8A*).

In our model, Pan1 and End3 recruit Sla1 (*Sun et al., 2015*), and the Pan1-End3-Sla1 complex guides endocytic actin assembly to endocytic sites through an SH3-PRM interaction network. The metazoan counterpart of Pan1-End3-Sla1 is likely intersectin (ITSN), which has both EH domains and SH3 domains in a single protein (*Tsyba et al., 2011*). ITSN was reported to interact with N-WASP and to regulate actin assembly (*Hussain et al., 2001*; *McGavin et al., 2001*). Furthermore, recent studies suggest that the vaccinia virus protein A36 recruits ITSN1 to the virus prior to its actin-based motility, and that ITSN1 promotes N-WASP-dependent actin polymerization (*Donnelly et al., 2013*; *Humphries et al., 2014*; *Snetkov et al., 2016*). Thus, the Pan1-End3-Sla1 function revealed here may reflect a general role of ITSN in spatiotemporally regulating N-WASP-dependent actin polymerization in various cellular processes.

## WASP and WIP play central roles in recruiting the actin machinery via a robust, multivalent SH3 domain-PRM interaction network at endocytic sites

Here, we developed a novel 'end3C fusion' method to recruit proteins to endocytic sites independent of the Sla1 SH3 domains and the Pan1 PRD. One important feature of this method is that the end3C fusion does not affect the native expression levels of the proteins being tagged. Underproduction or overproduction of Las17 or Vrp1 tends to dramatically influence their cellular functions (*Shively et al., 2013*), likely complicating interpretations. Using this powerful in vivo system, we were able to demonstrate that Las17 and Vrp1 preferentially interact with each other, and together

they are necessary and sufficient to recruit the remaining WASP/Myosin module components to endocytic sites in *sla1W41AW108A pan1ΔPRD* cells.

Several lines of evidence support the conclusion that once they are recruited by the multivalent Pan1-End3-Sla1 complex, Las17 and Vrp1 in turn recruit their binding partners to endocytic sites through additional multivalent PRM and SH3-domain interactions, greatly expanding the interaction network. The Las17 PRD (proline-rich domain) contains 20 SH3-binding PRMs (proline-rich motifs) (*Feliciano and Di Pietro, 2012*). Vrp1 is very rich in proline and also contains more than 20 PRMs (*Donnelly et al., 1993*). In budding yeast, there are only 25 SH3 domain-containing proteins in total. Remarkably, 11 of them interact with Vrp1 and/or Las17 through PRM-SH3 domain interactions (*Anderson et al., 1998*; *Geli et al., 2000*; *Rajmohan et al., 2009*; *Tarassov et al., 2008*; *Tong et al., 2002*; *Tonikian et al., 2009*; *Verschueren et al., 2015*). These proteins arrive at endocytic sites with similar timing to, or after, Las17 and Vrp1 (*Figure 8A*)(*Boettner et al., 2011*). Some Vrp1- and Las17-binding proteins themselves have multiple SH3 domains, either in their primary sequences or as the result of dimerization or oligomerization. For example, Bzz1 contains two SH3 domains and the hetero-dimeric N-BAR protein (*Kishimoto et al., 2011*) Rvs161/167 dimerizes through N-BAR domains (*Colwill et al., 1999*). In addition, in vitro studies have shown that some of these SH3 proteins are able to bind to multiple PRMs of the Las17 PRD (*Tong et al., 2002*). Multiple dimeric and oligomeric SH3 proteins can interact in a complex network with the PRMs of Las17 and Vrp1, resulting in highly cooperative binding. The partial rescue of function when end3C is fused to Bzz1, Rvs167, or Myo5 (*Figure 5B*), supports the notion that these proteins promote interactions that concentrate key actin assembly factors at endocytic sites.

Cooperativity in binding of multivalent proteins would greatly increase the robustness of the network and could promote actin nucleation activity (discussed further in the following section), facilitating the rapid recruitment and activation of the endocytic actin machinery (*Figure 8A*). Interestingly, the proteins mentioned here are well conserved in mammals (*Cheng et al., 2012*; *Merrifield and Kaksonen, 2014*) and other organisms (*Lam et al., 2001*; *Sirotkin et al., 2005*; *Xin et al., 2013*), indicating that multivalent PRM-SH3 network formation centered around WASP and WIP may be a general feature in spatiotemporal control of Arp2/3 complex-mediated actin assembly.

## Evidence that a WASP- and WIP-centered multivalent SH3 domain-PRM network triggers actin assembly onset in a switch-like manner through a possible transient phase separation

Endocytic actin nucleation mainly depends on the yeast WASP (Las17) and the type 1 myosin-WIP complex (Myo3/5-Vrp1) (*Galletta et al., 2008*; *Lewellyn et al., 2015*; *Sirotkin et al., 2005*; *Sun et al., 2006*). Previous in vitro data suggest that Las17 NPF activity is constitutive (*Feliciano and Di Pietro, 2012*; *Rodal et al., 2003*). Vrp1 is required for Myo3/5 recruitment and NPF activation (*Sirotkin et al., 2005*; *Sun et al., 2006*). However, since Las17 and Vrp1 arrive at endocytic sites 15–20 s prior to actin assembly and Myo3/5 recruitment, the mechanism that controls the onset of Arp2/3 activation in vivo has been mysterious.

Our quantitative analysis of Las17 and Vrp1 recruitment in different genetic backgrounds provides several important new insights into NPF-mediated actin nucleation regulation by Las17 and Myo3/5-Vrp1 in vivo. Las17 NPF activity does not trigger immediate actin nucleation at endocytic sites even when we genetically disable Las17 inhibition by Sla1 in live cells (*Figure 5D*). Co-existence of Vrp1 and type I myosin at endocytic sites is not sufficient to induce actin assembly until Las17 and Vrp1 levels rise to an apparent threshold level (*Figure 7D*). Thus, in contrast to what was previously assumed based on in vitro studies (*Feliciano and Di Pietro, 2012*; *Rodal et al., 2003*; *Sirotkin et al., 2005*; *Sun et al., 2006*), the Las17 NPF and the Vrp1-dependent type I myosin NPF activities do not appear to be active when present at low concentrations. Importantly, our quantitative analysis strongly suggested that productive actin assembly initiation is tightly coupled to accumulation of a threshold concentration of Las17 and Vrp1 at endocytic sites (*Figure 5G–5I* and *Figure 8*). Furthermore, the actin assembly rate appears to be very similar irrespective of how and when Las17 and Vrp1 are recruited to endocytic sites at sufficient levels, and assembly appears to be 'all or nothing' (*Figure 6E*), leading us to propose a 'switch-like' activation mechanism.

Previous studies proposed a hierarchical model for N-WASP NPF activation: allostery and dimerization, which control accessibility and affinity of the N-WASP VCA for the Arp2/3 complex, respectively (*Higgs and Pollard, 2000*; *Padrick et al., 2008*; *Padrick and Rosen, 2010*; *Rivera et al.,*

*2009*). Our results provide support for a similar mechanism in cells. Las17 does not appear to contain a G protein binding domain, and its NPF activity is not auto-inhibited. Therefore, Las17 does not depend on allosteric control to gain 'basal' NPF activity, explaining previous in vitro results (*Feliciano and Di Pietro, 2012*; *Rodal et al., 2003*). However, our results suggest that the 'basal' Las17 NPF activity is not sufficient to trigger actin assembly unless Las17 and Vrp1 are concentrated to a threshold level in vivo. We suggest that multivalent PRM and SH3-domain interactions between Las17, Vrp1 and their binding partners (discussed in the previous section), at endocytic sites induces formation of a higher-order complex, in which two or more VCAs are brought together to enhance affinity for the Arp2/3 complex. Consistently, previous studies using whole cell exacts found that Las17 is part of a large and biochemically stable complex (*Feliciano and Di Pietro, 2012*; *Soulard et al., 2002*). In our proposed scenario, Las17 and Vrp1 accumulate at endocytic sties to a threshold level and their PRMs provide a high local concentration of multivalent interactions through which Myo3 and Myo5 are also recruited and activated in an 'all or nothing' manner. Thus, Las17 NPF activation and Vrp1-dependent Myo3/5 recruitment and NPF activation are triggered simultaneously, creating a burst of actin filament assembly, upon which the Myo3/5 motor domains can exert forces, collectively generating forces required for endocytic membrane invagination and membrane scission.

The relationship between Las17 and Vrp1 recruitment and the switch-like onset of actin assembly we observe in live cells is consistent with a mechanism based on in vitro studies in which multivalent SH3-domain and PRM interactions induce a phase transition centered around N-WASP to promote local actin assembly (*Banjade and Rosen, 2014*; *Li et al., 2012*). We speculate that the cytoplasmic WASP-Myosin module puncta propelled by actin comet tails observed when endocytic site formation is uncoupled from actin assembly (Pan1-End3 depleted cells [*Sun et al., 2015*] or *sla1W41AW108A pan1ΔPRD* cells [*Figure 4F*]) is triggered by multivalent SH3-domain and PRM interactions in the cytoplasm. Consistently, removing multi-PRM-containing proteins (Las17 or Vrp1) from the cytoplasm, completely suppressed cytoplasmic WASP-Myosin module puncta formation and assembly of associated actin comet tails (*Figure 5C–F*).

Overall, our results strongly support a model in which accumulation of WASP and WIP to a threshold level at endocytic sites establishes a robust, multivalent SH3 domain-PRM interaction network (possibly involving a transient phase separation [*Banjade and Rosen, 2014*; *Li et al., 2012*]), which triggers the onset of actin assembly in a switch-like manner in vivo. It is certainly possible that the other factors, such as lipids and other endocytic modules involved in endocytic site establishment or cargo loading, also influence the network and actin assembly onset. Future efforts need to assess how such factors cooperate with WASP and WIP to facilitate productive endocytic actin assembly.

## Materials and methods

### Media, strains and plasmids

Yeast strains were grown in standard rich media (YPD) or synthetic media (SD) supplemented with 0.2% Casamino acids. The yeast strains are listed in *Supplementary file 1*. GFP, mCherry, RFP, TagRFP-T and end3C tags were integrated at the C-terminus of each gene as described previously (*Lee et al., 2013*; *Longtine et al., 1998*).

### Fluorescence microscopy and image analysis

Fluorescence microscopy was performed using a Nikon Eclipse Ti microscope (Nikon Instruments, Melville, NY) controlled by Metamorph (Molecular Devices, Sunnyvale, CA), equipped with a Plan Apo VC 100×/1.4 Oil OFN25 DIC N2 objective (with Type NF immersion oil, Nikon), a Perfect Focus System (Nikon), and a Neo sCMOS camera (Andor Technology Ltd., South Windsor, CT) (65 nm effective pixel size). For live cell imaging, cells were grown to early log phase at 25°C. The cells in synthetic media were adhered to the surface of a concanavalin A coated (0.1 µg/ml) coverslip. All imaging was done at room temperature. For single-channel live cell imaging, images were acquired continuously at 1 frames/sec. Two-channel movies were made using the SPECTRA X Light Engine (Lumencor, Beaverton, OR) for excitation with a 524/628 nm dual-band bandpass filter for GFP/mCherry emission (Brightline, Semrock, Lake Forest, IL). Time to acquire one image pair is 1.1 s, or 1.3 s, or 1.8 s depending on the signal intensity.

For multifocus imaging of Sac6-GFP labeled actin comet tails, images were acquired continuously at 4 frames/sec using multifocus microscopy (MFM) as described previously (*Abrahamsson et al., 2013*).

Image J software was used for general manipulation of images and movies, for preparing kymographs, and for data analysis and quantification. For the detailed analysis procedure, please see *Figure 1—figure supplement 2* and *Figure 2—figure supplement 1*. For patch lifetimes, unless otherwise stated, more than 100 patches were measured for each variant. Patches that at any point in their lifetime were too close to another patch to be clearly resolved were excluded from our analysis. For the fluorescence intensity profile, at least 10 patches were measured for each variant. These sample sizes are more than what has been used in other landmark papers in this type of study (*Bradford et al., 2015*; *Kaksonen et al., 2003*; *Newpher et al., 2005*). For each pair of variables, pooled data of analysis were compared by a two sided Mann-Whitney test using the Prism 7 graphing software.

### Lucifer Yellow uptake assay

Lucifer Yellow uptake assay was also done as previously described (*Sun et al., 2006*). Cells were grown to early log phase in YPD media. Approximately $1 \times 10^7$ cells were pelleted and resuspended in 90 µl of YPD and 10 µl of 40 mg/ml Lucifer yellow CH dilithium salt. Cells were incubated for 90 min at room temperature and then washed four times in ice-cold 50 mM potassium phosphate buffer, pH 7.4, containing 10 mM $NaN_3$ and 10 mM NaF. Cells were then imaged at room temperature.

## Acknowledgements

We thank the members in Drubin/Barnes laboratory for helpful discussions. We are grateful to Matt Akamatsu for advice on image analysis and to Jonathan Wong for critically reading the manuscript. We also thank Mustafa Mir and Sara Abrahamsson for their help in implementing MFM microscopy. This work was supported by NIH grant R35GM118149 to DGD.

## Additional information

### Funding

| Funder | Grant reference number | Author |
| --- | --- | --- |
| National Institutes of Health | R35GM118149 | David G. Drubin |

The funders had no role in study design, data collection and interpretation, or the decision to submit the work for publication.

### Author contributions

YS, Conceptualization, Formal analysis, Investigation, Methodology, Writing—original draft, Project administration, Acquisition of data; NTL, TJ, Investigation, Acquisition of data; AT, Methodology, Acquisition of data; XD, Resources, Supervision, Methodology; DGD, Resources, Supervision, Funding acquisition, Project administration, Writing—review and editing

### Author ORCIDs

Yidi Sun, http://orcid.org/0000-0002-2157-1983
Astou Tangara, http://orcid.org/0000-0001-5681-233X
Xavier Darzacq, http://orcid.org/0000-0003-2537-8395
David G Drubin, http://orcid.org/0000-0003-3002-6271

## Additional files

### Supplementary files

• Supplementary file 1. Yeast strains used in this study.

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
