## [Decision Letter]

[Editors’ note: a previous version of this study was rejected after peer review, but the authors submitted for reconsideration. The first decision letter after peer review is shown below.]

Thank you for submitting your work entitled "WASP recruitment to a threshold level by multivalent linker proteins leads to switch-like Arp2/3 activation in vivo" for consideration by *eLife*. Your article has been reviewed by three peer reviewers, one of whom, (Pekka Lappalainen) (Reviewer #1) is a member of our Board of Reviewing Editors and a Senior Editor.

Our decision has been reached after consultation between the reviewers. Based on these discussions and the individual reviews below, we regret to inform you that your work will not be considered further for publication in *eLife*.

All three reviewers found the data of very good technical quality and stated that your study provides interesting new insights into the protein network that recruits nucleation-promoting factors to the site of endocytosis. However, they also felt that the most important and novel conclusion of the study, concerning the dependence of actin filament assembly in an 'all or nothing' manner on Las17/WASP accumulation beyond a threshold level, is not sufficiently well supported by the data presented. Thus, an extensive amount of additional work would be required to address this point. Because our policy is that a revision is invited only when it can be carried out in two-three months, we cannot offer to publish this work in *eLife*.

Reviewer #1 and #3, however, provide some suggestions for testing this hypothesis e.g. by using a mutant version of Myo3/5, and/or by manipulating the Las17 expression levels in ectopic structures in a Sla1W41AW108A:pan1delPRD background. There are certainly also other ways to test the hypothesis by making use of various genetic approaches and mutant versions of nucleation-promoting factors. Therefore, if you can provide much stronger support for the 'threshold hypothesis' or reveal an alternative mechanism responsible for the switch-like behavior of actin assembly onset, we would be glad to consider a new submission on this topic for publication in *eLife*. In this case, the new submission would be most likely evaluated by the three original reviewers.

Reviewer #1:

The mechanisms by which budding yeast WASP (Las17) and type I myosin (Myo3/5) nucleation-promoting factors (NPFs), together with a large array of scaffolding proteins, induce actin filament assembly at the sites of endocytosis are incompletely understood. Here, Sun et al., demonstrate that SH3 and poly-proline domain containing scaffolding proteins have overlapping roles in recruiting NPFs to the sites of endocytosis, and that the requirement of several linker proteins can be bypassed by expression of Las17 or Vrp1 fused to the C-terminal domain of the End3 (and thus artificially recruiting these proteins to the sites of endocytosis). Finally, the authors provide evidence that actin filament assembly is initiated only after Las17 has accumulated beyond a threshold level.

This study provides interesting new insights into the protein interplay regulating actin filament assembly during endocytosis. Especially the experiments demonstrating that endocytic proline-rich and SH3 domain proteins generate a robust interaction network to orchestrate actin assembly are interesting and convincing. However, in my opinion additional experiments are required to test the hypothesis where 'WASP threshold level' provides a switch-like behavior for actin assembly onset at the sites of endocytosis.

1) The authors propose a model where actin filament assembly in an 'all or nothing' manner is dependent on Las17 accumulation beyond a threshold level (and propose that releasing Las17 from an inhibitory complex does not induce rapid actin filament polymerization at the late stages of endocytosis). However, the authors should provide stronger evidence for this model and/or consider and test alternative hypotheses for the timing of actin assembly onset. From the data presented in the manuscript and earlier publications, it seems that also Myo3/5 recruitment coincides with rapid actin assembly at the sites of endocytosis. Would it be possible that Myo3/5 activates Las17? This could be studied e.g. by using a mutant version of Myo3/5, where the NPF activity is abolished. The authors could for example test whether the NPF activity of Myo3/5 is essential to trigger actin assembly in sla1W41AW108A background (if this is indeed the case, it would provide additional support for the model presented by the authors). Moreover, one could examine whether a Myo3/5 NPF mutant displays genetic interactions with sla1W41AW108A. Also any information concerning the mechanism by which Myo3/5 is rapidly recruited to the sites of endocytosis at the stage when actin assembly is triggered would significantly improve this study.

2) In my opinion, the genetic data presented in the manuscript implies that Pan1 does not interact with Sla1 in cells, but instead suggest that End3 interacts with Sla1 and recruits Sla1 and Las17 to the sites of endocytosis. This is also supported by the data presented in Figure 5 demonstrating that the Sla1 function in recruitment of Las17 can be bypassed by expressing Las17-end3C fusion protein. This could perhaps be clarified in Figure 7.

Reviewer #2:

Using live cell imaging and quantitative analysis of a series of Sla and Pan1 mutants together with a number of GFP/RFP tagged markers (Las17, Vrp1, Myo5 and Abp1) the authors further explore the mechanisms regulating Las17-dependent actin polymerization during yeast endocytosis. The authors uncover that the first two SH3 domains of Sla1 and the PRD of Pan1 are not required for their recruitment to endocytic sites but are essential for productive Las17-dependent actin polymerization during endocytosis. The authors provide evidence that the first two SH3 domains of Sla1 are required for the correct temporal recruitment of Las17 to ensure actin polymerization is coupled to endocytosis. Their quantitative analysis also suggests that the PRD of Pan1 and additional proteins contributes to the correct temporal recruitment of Las17. In the absence of the first two SH3 domains of Sla1 and the PRD of Pan1 the Las17-Myosin module proteins are not recruited to endocytic sites and actin assembly is uncoupled from endocytosis. Using a Las17-end3C hybrid the authors show that artificially directing Las17 to endocytic sites independently SH3 and PRD motifs is sufficient to rescue endocytosis and cell growth. Moreover, by performing quantitative two color imaging of endocytic patches the authors show that actin polymerization (recruitment of Abp1-RFP) is initiated once Las17 reaches a threshold of 70-80% of its maximum value. Based on this the authors suggest Las17 regulates actin nucleation during endocytosis in a switch like manner.

Overall the data and analysis is very thorough and supports the authors' conclusions although I'm not sure I would say this is a switch per se. A similar dose response in promoting N-WASP dependent actin nucleation also appears to be operating in other systems including artificial Nck clusters (see papers from Bruce Mayer lab) and Vaccinia actin tail formation (Figure 3 Humphries et al., JCS 2014). This should probably be mentioned.

In subsection “A WASP-end3C chimeric protein restores normal growth when SH3 and proline rich domains of multivalent endocytic linker proteins are absent.” the authors say that Las17 central PRD domain contains 20 potential SH3 binding sites. A similar situation exists for N-WASP and yet only two sites are capable of binding the three SH3 domains of Nck (Donnelly et al., Current Biology 2013). Moreover, the same is true for WIP (Vrp1). Furthermore, N-WASP is not recruited through interaction of its PRD domain with the SH3 domain of Nck but by virtue of its association with WIP. It is also possible to recruit N-WASP to Vaccinia virus without inducing its activationand actin polymerization. These data using Vaccinia would suggest that SH3/PRD interactions in protein networks in cells maybe more specific and constrained than the number of predicted binding sites would suggest. It is worth also mentioning that many of the in vitro assays looking at multivalent SH3/PRM interactions also lack spatial constraints such as would be occurring in yeast actin patches. Random associations between multivalent SH3/PRM motifs would presumably not result in a switch like behaviour and the recruitment of Abp1 when Las17 is at 70-80% of its maximum intensity, but a broad range of values over different time scales. The authors might like to comment on this?

It is striking that Vrp1-end3C is as effective as Las17-end3C in rescuing actin polymerization and endocytosis in the absence of SH3 and PRD motifs (Figure 5). This raises the question whether Sla1 and Pan1 are interacting with Vrp1 to recruit Las17 to endocytic sites or whether Las17 is recruited directly and Vrp1 comes along for the ride. The authors should provide temporal information of the patch fluorescent intensities for Vrp1-end3C (as shown in Figure 5 for Las17-end3C, which is currently down as unpublished in discussion). It might also be possible using the right mutants of Las17/Vrp1 to determine who is recruiting who in the system.

Reviewer #3:

The manuscript presented by Yidi Sun and coworkers describes a collection of nicely performed experiments with clear results, showing that, 1) combined mutation of the SH3 domains of Sla1 and the poly proline domain (PRD) of Pan1 completely uncouple actin polymerization by the endocytic WASP/Myo module from the endocytic coat, and that, 2) the strong endocytic defects installed in the sla1W41AW108A pan1∆PRD mutant can be rescue by fusion proteins of the End3 C-terminal domain with either the yeast WASP or WIP homologues Las17 or Vrp1, respectively. Even though the experiments are well performed, these results do not represent a major advance in the field with respect to what the same authors or others already published. Thus, for example, the group of D. Drubin published in 2015 an interesting paper where they demonstrated that depletion of End3 and Pan1 causes uncoupling of actin polymerization from the endocytic coat, by a failure to recruit Sla1. Also, the role of the Sla1 SH3 domains recruiting Las17 has already been described by the group of Di Pietro, and the interaction between the PRD of Pan1 and the other major endocytic actin nucleating promoting factor (Myo5) has been described by the group of B. Wendland.

Probably being aware of this fact, the authors center the main conclusion of the paper (evident in the title) on a model that proposes that oligomerization of the WASP/Myo module at endocytic sites to a certain threshold level is the trigger for actin polymerization, as opposed to a mechanism where Las17 and Myo5 activities are modulated by other endocytic proteins, lipids or posttranslational modifications. This conclusion is mostly based in Figure 6, which shows that actin polymerization at endocytic sites is triggered when the Las17 signal reaches about 80% of its maximum intensity both in wild type cells expressing Las17-GFP or in cells expressing the Las17-End3C-GFP in the sla1W41AW108A pan1∆PRD background.

This conclusion is very interesting but also very preliminary and would certainly need reinforcement by showing a tight correlation between the Las17 threshold and actin polymerization. The authors could for example test if direct linking of Las17 to the early module components (Ede1, Apl1, clathrin…..) initiates actin polymerization before cargo loading. Most convincingly, the authors could artificially recruit Las17 to an ectopic structure with an inducible system, to show that actin polymerization is triggered at the same threshold as in endocytic sites. If this is the case, the experiment would discard a direct modulation of the Las17 or Myo5 activities by other endocytic coat components in the sla1W41AW108A pan1∆PRD background.

Experiments showing oligomerization of Las17 in vivo would also strongly reinforce the model.

[Editors’ note: what now follows is the decision letter after the authors submitted for further consideration.]

Thank you for submitting your article "Switch-like Arp2/3 activation upon WASP and WIP recruitment to a threshold level by multivalent linker proteins in vivo" for consideration by *eLife*. Your article has been reviewed by three peer reviewers, one of whom, Pekka Lappalainen (Reviewer #1) is a member of our Board of Reviewing Editors and Anna Akhmanova as the Senior Editor. The following individual involved in review of your submission have agreed to reveal their identity: Michael Way (Reviewer #2).

The reviewers have discussed the reviews with one another and the Reviewing Editor has drafted this decision to help you prepare a revised submission.

Summary:

This study provides interesting new insights into the interactions between Pan1, Sla1, WASP (Las17), and WIP (Vrp1) as well as their effects on actin assembly in yeast cells. Sun et al., demonstrate that functional SH3 domains of Sla1 and the proline-rich domain of Pan1 are critical for recruiting the WASP/Myosin-1/WIP actin polymerization module to the sites of endocytosis, and that disruption of these interactions results in formation of cytoplasmic, non-productive actin comet tails. Moreover, they show that restoring the WASP or WIP localization to the sites of endocytosis using various engineered fusion proteins (where either WASP, WIP or selected endocytic SH3/proline-rich proteins were fused to the C-terminal region of End3) can redirect actin assembly to endocytic sites in cells lacking the functional SH3 domains of Sla1 and proline-rich domain of Pan1. Finally, the new version of manuscript also provides evidence that actin assembly at endocytic sites may be triggered upon recruitment of WASP and WIP to a threshold level due to multivalent SH3 domain – polyproline interactions.

Although all three reviewers found the manuscript significantly improved, they felt that an additional control experiment and significant revision of the text are required before this manuscript is acceptable for publication.

Essential revisions:

1) This manuscript demonstrates that initiation of actin assembly at the sites of endocytosis correlates well with the WIP/WASP levels (in wild-type and mutant yeast strains). However, this is only a correlation and the study does not provide definitive proof that accumulation of WASP/WIP above the 'threshold' level indeed is solely responsible for actin assembly. Therefore, the authors should tone down this conclusion in the 'abstract' and 'discussion', and acknowledge that although their data support the WASP/WIP threshold model, also alternative/additional mechanisms may contribute the switch-like behavior of actin assembly at endocytic sites.

2) The authors should better explain the rationale of Myo5-End3C experiment presented in Figure 7. For clarity, they should also examine whether the NPF activity is retained in the Myo5-ENd3C protein (this could be tested by expressing it in a myo3D/myo5D/las17-WCAD strain to check functionality of this particular biochemical activity in vivo).

---

## [Author Response]

[Editors’ note: the author responses to the first round of peer review follow.]

*All three reviewers found the data of very good technical quality and stated that your study provides interesting new insights into the protein network that recruits nucleation-promoting factors to the site of endocytosis. However, they also felt that the most important and novel conclusion of the study, concerning the dependence of actin filament assembly in an 'all or nothing' manner on Las17/WASP accumulation beyond a threshold level, is not sufficiently well supported by the data presented. Thus, an extensive amount of additional work would be required to address this point. Because our policy is that a revision is invited only when it can be carried out in two-three months, we cannot offer to publish this work in eLife.*

*Reviewer #1 and #3, however, provide some suggestions for testing this hypothesis e.g. by using a mutant version of Myo3/5, and/or by manipulating the Las17 expression levels in ectopic structures in a Sla1W41AW108A:pan1delPRD background. There are certainly also other ways to test the hypothesis by making use of various genetic approaches and mutant versions of nucleation-promoting factors. Therefore, if you can provide much stronger support for the 'threshold hypothesis' or reveal an alternative mechanism responsible for the switch-like behavior of actin assembly onset, we would be glad to consider a new submission on this topic for publication in eLife. In this case, the new submission would be most likely evaluated by the three original reviewers.*

*Reviewer #1:*

*The mechanisms by which budding yeast WASP (Las17) and type I myosin (Myo3/5) nucleation-promoting factors (NPFs), together with a large array of scaffolding proteins, induce actin filament assembly at the sites of endocytosis are incompletely understood. Here, Sun et al., demonstrate that SH3 and poly-proline domain containing scaffolding proteins have overlapping roles in recruiting NPFs to the sites of endocytosis, and that the requirement of several linker proteins can be bypassed by expression of Las17 or Vrp1 fused to the C-terminal domain of the End3 (and thus artificially recruiting these proteins to the sites of endocytosis). Finally, the authors provide evidence that actin filament assembly is initiated only after Las17 has accumulated beyond a threshold level.*

*This study provides interesting new insights into the protein interplay regulating actin filament assembly during endocytosis. Especially the experiments demonstrating that endocytic proline-rich and SH3 domain proteins generate a robust interaction network to orchestrate actin assembly are interesting and convincing. However, in my opinion additional experiments are required to test the hypothesis where 'WASP threshold level' provides a switch-like behavior for actin assembly onset at the sites of endocytosis.*

We thank Dr. Lappalainen for finding that our work is interesting and convincing. In our revised manuscript, we have followed your advice and have added substantial amounts of new data to further support our proposed model.

*1) The authors propose a model where actin filament assembly in an 'all or nothing' manner is dependent on Las17 accumulation beyond a threshold level (and propose that releasing Las17 from an inhibitory complex does not induce rapid actin filament polymerization at the late stages of endocytosis). However, the authors should provide stronger evidence for this model and/or consider and test alternative hypotheses for the timing of actin assembly onset. From the data presented in the manuscript and earlier publications, it seems that also Myo3/5 recruitment coincides with rapid actin assembly at the sites of endocytosis. Would it be possible that Myo3/5 activates Las17? This could be studied e.g. by using a mutant version of Myo3/5, where the NPF activity is abolished. The authors could for example test whether the NPF activity of Myo3/5 is essential to trigger actin assembly in sla1W41AW108A background (if this is indeed the case, it would provide additional support for the model presented by the authors). Moreover, one could examine whether a Myo3/5 NPF mutant displays genetic interactions with sla1W41AW108A. Also any information concerning the mechanism by which Myo3/5 is rapidly recruited to the sites of endocytosis at the stage when actin assembly is triggered would significantly improve this study.*

In addition to Las17, Myo3/5 is another major NPF, which requires Vrp1 (yeast WIP) for its recruitment and NPF activity. In the revised manuscript, we addressed how Vrp1-dependent Myo3/5 NPF activity is involved in the onset of endocytic actin assembly. Our new data show that when cortical localization of either Las17 or Vrp1 is restored through an end3C fusion, the other appears to get recruited simultaneously and restore normal cell growth and productive endocytic actin assembly in *sla1W41AW108A pan1ΔPRD* cells (Figure 5 and Figure 6). In *sla1W41AW108A pan1ΔPRD* cells, Las17-end3C-GFP and Vrp1-mCherry, or Vrp1-end3C-GFP and Las17-TagRFP-T, respectively, develop fluorescence intensity together with identical kinetics at cortical patches (Figure 6 and Figure 6—figure supplement 1). In addition, we provided additional quantitative analysis (Figure 5) to demonstrate that accumulation of both Las17 and Vrp1 at endocytic sites to an apparent threshold level is tightly coupled to the onset of actin assembly in wild-type cells and in *sla1W41AW108A pan1ΔPRD* cells. Thus, our results suggest that a threshold accumulation of both Las17 and Vrp1 predicts the onset of actin nucleation, as well as Myo3/5 recruitment, in a switch-like manner. In other words, our new results suggest that the threshold level of Vrp1 triggers Myo3/5 recruitment in a switch-like manner. Moreover, our genetic studies showed that Vrp1 or Las17 is required for Las17-end3C or Vrp1-end3C, respectively, to restore normal growth of *sla1W41AW108A pan1ΔPRD* (Figure 6). In *sla1W41AW108A pan1ΔPRD* cells, in which scaffolding protein interactions have been surgically eliminated, (artificial) recruitment of Las17 and Vrp1 to endocytic sites is necessary and sufficient to restore productive endocytic actin assembly at cortical endocytic sites upon reaching a threshold level.

Based on previous studies, Myo3/5 do not appear to be required for activating Las17’s NPF activity. This is because the timing of the actin assembly is not affected in the *myo3Δ myo5Δ* cells or in *myo5-CAΔ myo3Δ* cells (Sun et al., 2006 Figure 3). In addition, *sla1Δ* does not show a synthetic interaction with *myo5CAΔmyo3Δ* (Figure 1—figure supplement 1), and actin assembly still occurs in an *sla1Δ myo5CAΔmyo3Δ* mutant (our unpublished data), further indicating that Myo3/5 NPF activity is not required for actin assembly by Las17 in the absence of Sla1.

Most significantly, we altered the arrival kinetics of Myo5 at endocytic sites by fusing Myo5 to end3C (Figure 7). Myo5-end3C-GFP is recruited (through end3C-Pan1 interaction) to endocytic sites much earlier with a longer lifetime, which is similar to Las17 and Vrp1 (Figure 7). However, in sharp contrast to the situation in wild-type cells, actin assembly is not triggered coincident with Myo5 recruitment or during the first 12.6 ± 4.9 sec of the Myo5-end3CGFP lifetime (Figure 7), even though Myo5-end3C and Vrp1 (as well as Las17) are all present at cortical patches (Figure 7). Instead, the onset of actin assembly still occurred when Las17 and Vrp1 accumulated to the levels at which actin assembly is observed in wild-type cells. These data using engineered variants with altered recruitment timing show that the arrival of Myo3/5 at endocytic sites is not sufficient to activate Las17 NPF activity and instead strongly support a model in which recruitment of Las17 and Vrp1 to a threshold level triggers actin assembly.

We propose that in wild-type cells Las17 and Vrp1 are recruited to a threshold level via a robust interaction network involving PRMs and SH3 domains, achieving a high local concentration of the key actin-nucleating proteins. Once recruited to a threshold level, Las17 NPF activation, Vrp1-dependent Myo3/5 NPF activation, and Myo3/5 motor activity collectively generate forces required for endocytic membrane invagination and membrane scission.

*2) In my opinion, the genetic data presented in the manuscript implies that Pan1 does not interact with Sla1 in cells, but instead suggest that End3 interacts with Sla1 and recruits Sla1 and Las17 to the sites of endocytosis. This is also supported by the data presented in Figure 5 demonstrating that the Sla1 function in recruitment of Las17 can be bypassed by expressing Las17-end3C fusion protein. This could perhaps be clarified in Figure 7.*

Previous studies from other labs and our lab established that Pan1 and End3 coexist in a stable complex, that Sla1 is recruited to endocytic sites through this Pan1-End3 complex and that End3 plays a crucial role through interactions of its N-terminus (Boeke et al., 2014; Sun et al., 2015). Meanwhile, Pan1 plays an apparently redundant role in Sla1 recruitment. Thus, Pan1, End3, and Sla1 function as a complex at the endocytic sites. We have modified the text and model figure to provide more clarity about what has been established in the field based on evidence from multiple labs.

*Reviewer #2:*

*Using live cell imaging and quantitative analysis of a series of Sla and Pan1 mutants together with a number of GFP/RFP tagged markers (Las17, Vrp1, Myo5 and Abp1) the authors further explore the mechanisms regulating Las17-dependent actin polymerization during yeast endocytosis. The authors uncover that the first two SH3 domains of Sla1 and the PRD of Pan1 are not required for their recruitment to endocytic sites but are essential for productive Las17-dependent actin polymerization during endocytosis. The authors provide evidence that the first two SH3 domains of Sla1 are required for the correct temporal recruitment of Las17 to ensure actin polymerization is coupled to endocytosis. Their quantitative analysis also suggests that the PRD of Pan1 and additional proteins contributes to the correct temporal recruitment of Las17. In the absence of the first two SH3 domains of Sla1 and the PRD of Pan1 the Las17-Myosin module proteins are not recruited to endocytic sites and actin assembly is uncoupled from endocytosis. Using a Las17-end3C hybrid the authors show that artificially directing Las17 to endocytic sites independently SH3 and PRD motifs is sufficient to rescue endocytosis and cell growth. Moreover, by performing quantitative two color imaging of endocytic patches the authors show that actin polymerization (recruitment of Abp1-RFP) is initiated once Las17 reaches a threshold of 70-80% of its maximum value. Based on this the authors suggest Las17 regulates actin nucleation during endocytosis in a switch like manner.*

*Overall the data and analysis is very thorough and supports the authors' conclusions although I'm not sure I would say this is a switch per se. A similar dose response in promoting N-WASP dependent actin nucleation also appears to be operating in other systems including artificial Nck clusters (see papers from Bruce Mayer lab) and Vaccinia actin tail formation (Figure 3 Humphries et al., JCS 2014). This should probably be mentioned.*

We thank Reviewer #2 for finding that our work is very thorough and supports our conclusions.

We regret that we did not do a better job referencing previous relevant research comprehensively. We hope to have fixed the problem in this revised manuscript.

Our quantitative analysis now demonstrates that actin assembly initiation is tightly coupled to an apparent threshold level of the two proteins Las17 and Vrp1 (Figure 5 and Figure 6), yeast WASP and WIP. Furthermore, once Las17 and Vrp1 reach to the threshold level in wild-type cells or in the *sla1W41AW108A pan1ΔPRD* mutant, actin assembly is induced in a very similar manner, with similar kinetics (Figure 6). Thus, we conclude that the onset of actin assembly is not only dependent on Las17 and Vrp1 reaching a threshold level, but also that it occurs in an “all or nothing” manner.

*In subsection “A WASP-end3C chimeric protein restores normal growth when SH3 and proline rich domains of multivalent endocytic linker proteins are absent.” the authors say that Las17 central PRD domain contains 20 potential SH3 binding sites. A similar situation exists for N-WASP and yet only two sites are capable of binding the three SH3 domains of Nck (Donnelly et al., Current Biology 2013). Moreover, the same is true for WIP (Vrp1). Furthermore, N-WASP is not recruited through interaction of its PRD domain with the SH3 domain of Nck but by virtue of its association with WIP. It is also possible to recruit N-WASP to Vaccinia virus without inducing its activation and actin polymerization. These data using Vaccinia would suggest that SH3/PRD interactions in protein networks in cells maybe more specific and constrained than the number of predicted binding sites would suggest. It is worth also mentioning that many of the* in vitro *assays looking at multivalent SH3/PRM interactions also lack spatial constraints such as would be occurring in yeast actin patches. Random associations between multivalent SH3/PRM motifs would presumably not result in a switch like behaviour and the recruitment of Abp1 when Las17 is at 70-80% of its maximum intensity, but a broad range of values over different time scales. The authors might like to comment on this?*

The reviewer raises excellent points. We agree that it is important to gain more detailed knowledge about the multivalent SH3-PRM interaction network generated by Las17 and Vrp1 and their binding partners at both molecular and biochemical level in future. However, some information about possible binding interactions is available. In budding yeast, five PRMs of Las17 have been shown to be important for binding to Sla1 (Feliciano et al., 2012). Sla1, but not Vrp1 (Sun et al., 2006), plays key roles in Las17 recruitment to endocytic sites (this work, Rodal et al., 2003; Feliciano et al., 2012). In addition, phage-display experiments (Tong et al., 2002) generated a map showing which Las17 PRMs are able to interact with numerous binding partners specifically. On the other hand, there is much less information about the Vrp1 PRD in yeast. Nevertheless, these data indicated that Las17 and Vrp1 might not behave exactly the same as NWASP and WIP in Vaccinia virus, although similar principles may apply. In addition, the endocytic proteins that appear during the endocytic initiation phases also contain multivalent domains such as EH domains, which potentially facilitate multivalent interactions. Interestingly, these early proteins show very irregular lifetimes until Pan1-End3-Sla1 appear. Future investigations are needed to address how the “endocytic initiation phase” interaction network may affect formation of the WASP-Myosin module network. Since a very detailed spatiotemporal pathway has been elucidated for each protein recruited to endocytic sites, we are hopeful that future studies using yeast will yield more information to address the issues raised here.

*It is striking that Vrp1-end3C is as effective as Las17-end3C in rescuing actin polymerization and endocytosis in the absence of SH3 and PRD motifs (Figure 5). This raises the question whether Sla1 and Pan1 are interacting with Vrp1 to recruit Las17 to endocytic sites or whether Las17 is recruited directly and Vrp1 comes along for the ride. The authors should provide temporal information of the patch fluorescent intensities for Vrp1-end3C (as shown in Figure 5 for Las17-end3C, which is currently down as unpublished in discussion). It might also be possible using the right mutants of Las17/Vrp1 to determine who is recruiting who in the system.*

We fully agree that the rescue by Vrp1-end3C is quite striking and has the potential to provide additional insights into timing and recruitment mechanisms. Therefore, in the revised manuscript, we added a substantial amount of new data to investigate how the engineered Vrp1-end3C protein rescues the endocytic defect of the *sla1W41AW108A pan1ΔPRD* mutant. Our new data shows that when restoring Las17 or Vrp1 cortical localization through an end3C fusion, the other protein gets recruited simultaneously, and restores normal cell growth and productive endocytic actin assembly in *sla1W41AW108A pan1ΔPRD* cells (Figure 5 and Figure 6). In *sla1W41AW108A pan1ΔPRD* cells, Las17-end3C-GFP and Vrp1-mCherry, or Vrp1-end3C-GFP and Las17-TagRFP-T, respectively, develop fluorescence intensity with indistinguishable kinetics at cortical patches (Figure 6 and Figure 6—figure supplement 1). Thus, Las17 and Vrp1 preferentially interact with each other, and these two proteins are necessary and sufficient to organize the remaining WASP/Myosin module components at endocytic sites for productive endocytosis in the *sla1W41AW108A pan1ΔPRD* mutant. This analysis also supports the conclusion that a threshold accumulation of both Las17 and Vrp1 is coupled to the switchlike onset of actin nucleation. Previous studies from others and our lab found that Vrp1 appears at endocytic sites slightly after Las17 in wild-type cells, and that Vrp1 recruitment is dependent on Las17. In the future, it will be important to explore whether Sla1 also interacts with Vrp1 through PRM-SH3 domain interactions.

*Reviewer #3:*

*The manuscript presented by Yidi Sun and coworkers describes a collection of nicely performed experiments with clear results, showing that, 1) combined mutation of the SH3 domains of Sla1 and the poly proline domain (PRD) of Pan1 completely uncouple actin polymerization by the endocytic WASP/Myo module from the endocytic coat, and that, 2) the strong endocytic defects installed in the sla1W41AW108A pan1∆PRD mutant can be rescue by fusion proteins of the End3 C-terminal domain with either the yeast WASP or WIP homologues Las17 or Vrp1, respectively. Even though the experiments are well performed, these results do not represent a major advance in the field with respect to what the same authors or others already published. Thus, for example, the group of D. Drubin published in 2015 an interesting paper where they demonstrated that depletion of End3 and Pan1 causes uncoupling of actin polymerization from the endocytic coat, by a failure to recruit Sla1. Also, the role of the Sla1 SH3 domains recruiting Las17 has already been described by the group of Di Pietro, and the interaction between the PRD of Pan1 and the other major endocytic actin nucleating promoting factor (Myo5) has been described by the group of B. Wendland.*

We appreciate that Reviewer #3 found that our experiments are well performed and that our results are clear. However, we respectfully disagree with the Reviewer #3’s assessment regarding the significance of our results, given what we see as novel mechanistic insights revealed by these studies. Our previous work, which implicated Pan1, End3 and Sla1 as playing vital roles in coupling actin assembly to endoctyic sites, involved depletion of full-length Pan1 and End3 proteins from the cell. However, the mechanism governing the coupling between the endocytic machinery and the actin assembly machinery could not be addressed by such studies. This is because Pan1, End3 and Sla1 contain more than ten functional motifs that are involved in numerous protein-protein interactions with clathrin, receptors, adapters, NPFs etc. Thus, an important yet challenging task and priority was to identify the specific domain(s) required for linking actin assembly to endocytic sites. This knowledge, in turn, would provide insights into the mechanism by which the actin assembly machinery is recruited to and becomes active at endocytic sites. We point out that the same proteins, WASP, WIP and Myosin 1 are recruited to a number of different sites under different circumstances in cells to provide forces for a variety or processes. Thus, the principles learned from our careful, quantitative analysis are likely to apply very broadly.

In this current work, we revealed that in cells in which Pan1’s PRMs and the two Sla1 SH3-domains have been specifically eliminated by construction of appropriate mutants, the identical actin phenotype is observed as in cells depleted for both Pan1 and End3. This is an important new finding because it provides strong evidence that Pan1 and Sla1 recruit the actin assembly machinery to endocytic sites through a set of SH3-PRM interactions. Although the interactions of Pan1 and Myo5 or Las17 and Sla1 have been described previously, loss of any of these interactions alone does not cause significant defects in endocytic actin assembly in vivo. Previous in vitroexperiments on these interactions have been interpreted as suggesting that they regulate the NPF activity of Myo3/5-Vrp1 and Las17. The assumptions that Las17 is not auto-inhibited, and that Sla1 is required to keep Las17 inhibited at the early stage of endocytic internalization, have led researchers to search for factors that relieve the inhibition of Las17 by Sla1, but without much success. In contrast, our in vivoresults indicate that the yeast WASP Las17, even without interacting with Sla1, is not as active as was previously believed. Importantly, our rescue experiments utilizing a novel end3C fusion strategy clearly suggest that the interactions of Pan1 with Myo5 or Las17 with Sla1 primarily function in protein recruitment in vivo. These results provide an explanation for why the search for factors that relieve Sla1 inhibition of Las17 have failed. Thus, our findings provide new insights into the activation mechanism for endocytic actin assembly and identify a new set of questions, advancing the general understanding of NPF activation mechanisms in living cells. Therefore, the experiments shown in the first part of our paper are novel and represent important advances in the field. Furthermore, the first part of our study also lays a strong foundation for the second part of the paper investigating the parameters that matter for triggering rapid actin assembly in vivo.

*Probably being aware of this fact, the authors center the main conclusion of the paper (evident in the title) on a model that proposes that oligomerization of the WASP/Myo module at endocytic sites to a certain threshold level is the trigger for actin polymerization, as opposed to a mechanism where Las17 and Myo5 activities are modulated by other endocytic proteins, lipids or posttranslational modifications. This conclusion is mostly based in Figure 6, which shows that actin polymerization at endocytic sites is triggered when the Las17 signal reaches about 80% of its maximum intensity both in wild type cells expressing Las17-GFP or in cells expressing the Las17-End3C-GFP in the sla1W41AW108A pan1∆PRD background.*

In this study, we have addressed two important questions. The first part addresses how the WASP-Myosin module is concentrated at endocytic sites. We identified a novel synthetic genetic interaction between mutants of Sla1’s SH3 domains and Pan1’s PRD, and demonstrated that the actin assembly driven by proteins of the WASP-Myosin module is uncoupled from cortical endocytic sites, which are rendered nonproductive. We then developed a novel “end3C fusion” strategy to artificially recruit proteins to endocytic sites without altering expression levels of the protein being tagged. Combination of our novel “end3C-fusion” method with the newly identified SH3 domain and PRD mutants allowed us to demonstrate that the robust PRMSH3 interaction network is the key to recruitment of the actin assembly machinery to endocytic sites. The second question concerns the mechanism by which the onset of actin assembly is triggered. With a substantial number of new experiments now added to our previously reported experiments (discussed in the following section), our results provide several important new insights in NPF-mediated actin nucleation regulation by Las17 and Myo3/5-Vrp1 in vivo. Las17 NPF activity does not trigger immediate actin nucleation at endocytic sites even in the absence of the Sla1 inhibition in live cells (Figure 5). Co-existence of Vrp1 and type I myosin at endocytic sites is not sufficient to induce actin assembly until Las17 and Vrp1 levels rise to an apparent threshold level (Figure 7). Thus, in contrast to what was previously assumed based on in vitrostudies (Feliciano and Di Pietro, 2012; Rodal et al., 2003; Sirotkin et al., 2005; Sun et al., 2006), the Las17 NPF and the Vrp1-dependent type I myosin NPF do not appear to be active when present at low concentrations. Importantly, our quantitative analysis demonstrated that actin assembly initiation is tightly coupled to accumulation of a threshold concentration of Las17 and Vrp1 (Figure 5). Furthermore, the actin assembly rate appears to be very similar irrespective of how and when Las17 and Vrp1 are recruited to endocytic sites at sufficient levels, and assembly appears to be “all or nothing” (Figure 5 and Figure 6). Thus, Las17 NPF activation, Vrp1-dependent Myo3/5 NPF activation, and Myo3/5 motor activity are coincidentally recruited at sufficiently high local concentration to generate forces required for endocytic membrane invagination and membrane scission.

Finally, it is important to note that our proposed model is not exclusive of (or opposed to) the possibilities that the WASP-Myosin module might be modulated by additional factors (interaction with other endocytic proteins, lipids or posttranslational modifications) after they are recruited to endocytic sites, in order to coordinate activities and spatially organize components for successful endocytic internalization. Our current work is not aimed to test these possibilities. We now modified the title to avoid confusion. However, the formation of cytoplasmic actin comet tails when the link between the endocytic machinery and the actin assembly machinery is broken in *sla1W41AW108A pan1ΔPRD* does show that the WASP-Myosin module is sufficient to initiate and sustain actin assembly in the absence of the endocytic proteins that appear at the early endocytic stage.

*This conclusion is very interesting but also very preliminary and would certainly need reinforcement by showing a tight correlation between the Las17 threshold and actin polymerization.*

We appreciate that Reviewer #3 finds that our proposed model is interesting. We now followed Reviewer #3’s advice and generated new sets of data to further strengthen our model, summarized as following.

Our new data show that when cortical localization of either Las17 or Vrp1 is restored through an engineered end3C fusion, the other protein is recruited simultaneously and normal cell growth and productive endocytic actin assembly are robustly restored in *sla1W41AW108A pan1ΔPRD* cells (Figure 5 and Figure 6). In *sla1W41AW108A pan1ΔPRD* cells, Las17-end3C-GFP and Vrp1-mCherry, or Vrp1-end3C-GFP and Las17-TagRFP-T, respectively, develop fluorescence intensity with indistinguishable kinetics at cortical patches (Figure 6 and Figure 6—figure supplement 1). Furthermore, in the revised manuscript, we provided additional quantitative analysis (Figure 5) to demonstrate that accumulation of both Las17 and Vrp1 at endocytic sites to an apparent threshold level is tightly coupled to the onset of actin assembly in wild-type cells and in *sla1W41AW108A pan1ΔPRD* cells. Moreover, our genetic studies demonstrated that Vrp1 or Las17 is required for Las17-end3C or Vrp1-end3C, respectively, to restore normal growth to *sla1W41AW108A pan1ΔPRD* cells (Figure 6).

Collectively, our old and new results support a model in which threshold accumulation of both Las17 and Vrp1 is tightly coupled to the onset of actin nucleation as well as Myo3/5 recruitment and a burst of actin assembly, in a switch-like manner. In other words, our new results suggest that when a threshold level of Vrp1 (and Las17) is reached, Myo3/5 recruitment and actin assembly are triggered in a switch-like manner. Vrp1 is known from our previous studies and studies from other labs to be required for Myo3/5’s recruitment and NPF activity.

Thus, in *sla1W41AW108A pan1ΔPRD* cells, (artificial) recruitment of Las17 and Vrp1 are sufficient and necessary to restore productive endocytic actin assembly by Las17 NPF activity and Vrp1-dependent-Myo3/5 NPF activity, which are triggered in a switch-like manner upon Las17 and Vrp1 reaching a threshold level.

*The authors could for example test if direct linking of Las17 to the early module components (Ede1, Apl1, clathrin…..) initiates actin polymerization before cargo loading. Most convincingly, the authors could artificially recruit Las17 to an ectopic structure with an inducible system, to show that actin polymerization is triggered at the same threshold as in endocytic sites. If this is the case, the experiment would discard a direct modulation of the Las17 or Myo5 activities by other endocytic coat components in the sla1W41AW108A pan1∆PRD background.*

It could be technically very challenging to control/monitor local accumulation of Las17 in an inducible system that artificially recruits Las17 to an ectopic structure, but we do plan to study fusion of Las17 to early pathway proteins in a future study (see below). Instead, we altered the recruitment timing of Myo3/5 (Figure 7), which is one of the two major NPFs for endocytic actin assembly, to further test our model. As expected, Myo5-end3C-GFP is recruited (through end3C-Pan1 interaction) to endocytic sites much earlier and with a longer lifetime time (Figure 7), which is similar to Las17 and Vrp1 (Figure 7). However, actin assembly is not triggered during the first 12.6 ± 4.9 sec of Myo5-end3C-GFP lifetime (Figure 7), even though Myo5-end3C and Vrp1 (as well as Las17) are present at cortical patches (Figure 7). Remarkably, in cells expressing this engineered *MYO5-end3C-GFP* variant, even though the relative timing of WASP, WIP and Myosin 1 are altered, the onset of actin assembly still coincides with recruitment of a threshold level of Vrp1 (Figure 7 and Figure 7—figure supplement 1). Together, these data strongly support our model that threshold levels of Las17 and Vrp1 play a decisive role in inducing switch-like NFP activation through multivalent PRM and SH3-domain interactions in vivo.

The “Ede1-Las17 hybrid” idea is very interesting. However, as we mentioned above, our proposed model is not exclusive to the possibilities that the WASP-Myosin module may be modulated by other factors after they are recruited to endocytic sites in order to gain full activity for endocytic internalization. We do plan to generate Ede1-Las17 hybrid proteins to investigate these possibilities as a future study and hope to have convinced Reviewer #3 that our study is already very comprehensive and makes important advances in understanding WASP/WIP/Myosin 1-mediated actin assembly regulation in living cells.

*Experiments showing oligomerization of Las17* in vivo *would also strongly reinforce the model.*

Thanks for this suggestion. Indeed, previous studies using whole cell exacts showed that Las17 forms a large and biochemically stable complex (Feliciano et al., 2012), which is consistent with our model. We have now added this reference in support our model.

[Editors' note: the author responses to the re-review follow.]

*Essential revisions:*

*1) This manuscript demonstrates that initiation of actin assembly at the sites of endocytosis correlates well with the WIP/WASP levels (in wild-type and mutant yeast strains). However, this is only a correlation and the study does not provide definitive proof that accumulation of WASP/WIP above the 'threshold' level indeed is solely responsible for actin assembly. Therefore, the authors should tone down this conclusion in the 'abstract' and 'discussion', and acknowledge that although their data support the WASP/WIP threshold model, also alternative/additional mechanisms may contribute the switch-like behavior of actin assembly at endocytic sites.*

We agree that our data show a correlation between WASP/WIP levels and switch-like actin assembly onset, but do not definitively show causation. We apologize for writing the previous version of our manuscript in a manner that indicated that we had presented definitive evidence for causation. We have endeavored to tone down the conclusions in the Abstract and Discussion and to acknowledge the possibility that alternative mechanisms operate. Specifically, we modified the Abstract and added a paragraph at the end of the Discussion to address the reviewers’ concerns. We also made some minor changes in the Results section to summarize the conclusions in a more balanced manner.

To recap some of the main advances made in this study, we set out to determine how NPFs are concentrated at endocytic sites and how the actin assembly is triggered. Whereas excellent previous studies with similar goals were largely biochemical, our studies were performed using genetics and imaging in living cells for a natural process, endocytosis. Our results provide several important new insights: 1) We established that two Sla1 SH3 domains and the Pan1 PRD together play indispensable recruitment roles in coupling the WASP-Myosin machinery to endocytic sites. 2) We showed that the WASP-Myosin machinery is recruited to endocytic sites via SH3 domain-PRD interactions, and that WASP and WIP play central roles in this recruitment. 3) We showed that the yeast WASP NPF activity at endocytic sites does not appear to be regulated negatively by Sla1, or at least not exclusively in this manner, as was previously assumed. 4. We established through careful quantitative analyses that actin assembly and Type I Myosin recruitment/activation are triggered in an “all and nothing” manner.

The above results, considered in the context of results from recent in vitro studies of the Rosen lab, allow us to propose a mechanism in which WASP and WIP play central roles in establishing a robust multivalent SH3 domain-PRM interaction network at endocytic sites (likely through a transient phase separation), giving actin assembly onset a switch-like behavior in vivo. The tight correlation between reaching a threshold level (or a high level) of WASP/WIP, and actin assembly onset, provides additional strong evidenceto support our proposed model. This is because that formation of a robust multivalent SH3 domain-PRM interaction network may need WASP/WIP to attain a threshold concentration. It is possible that threshold levels of WASP and WIP are necessary but not sufficient to trigger productive endocytic actin assembly. We hope that more studies will follow ours to further examine our model and investigate how other factors cooperate with WASP and WIP to facilitate productive endocytic actin assembly.

*2) The authors should better explain the rationale of Myo5-End3C experiment presented in Figure 7. For clarity, they should also examine whether the NPF activity is retained in the Myo5-ENd3C protein (this could be tested by expressing it in a myo3D/myo5D/las17-WCAD strain to check functionality of this particular biochemical activity* in vivo).

We wrote an additional paragraph to explain why we performed the Myo5-end3C experiment (subsection “Recruitment and NPF activation of type I myosin by WIP appears to occur in a switch-like manne”).

We performed the suggested control experiment. *MYO5-end3C las17WCA∆ myo3∆* cells grow much better than *myo5CA∆ las17WCA∆ myo3∆* cells, suggesting that NPF activity is retained in Myo5-end3C. This new result is shown in Figure 5—figure supplement 2.